# Multi-Omic Advances in Olive Tree (*Olea europaea* subsp. *europaea* L.) Under Salinity: Stepping Towards ‘Smart Oliviculture’

**DOI:** 10.3390/biology14030287

**Published:** 2025-03-11

**Authors:** Manuel Gonzalo Claros, Amanda Bullones, Antonio Jesús Castro, Elena Lima-Cabello, María Ángeles Viruel, María Fernanda Suárez, Remedios Romero-Aranda, Noé Fernández-Pozo, Francisco J. Veredas, Andrés Belver, Juan de Dios Alché

**Affiliations:** 1Institute for Mediterranean and Subtropical Horticulture “La Mayora” (IHSM La Mayora-UMA-CSIC), 29010 Malaga, Spain; amandabullones@uma.es (A.B.); marian.viruel@eelm.csic.es (M.Á.V.); rromero@eelm.csic.es (R.R.-A.); noe.fernandez.pozo@csic.es (N.F.-P.); 2Department of Molecular Biology and Biochemistry, Universidad de Málaga, 29071 Malaga, Spain; fsuarez@uma.es; 3Department of Stress, Development and Signaling of Plants, Plant Reproductive Biology and Advanced Microscopy Laboratory (BReMAP), Estación Experimental del Zaidín, CSIC, 18008 Granada, Spain; antoniojesus.castro@eez.csic.es (A.J.C.); elena.lima@eez.csic.es (E.L.-C.); andres.belver@eez.csic.es (A.B.); juandedios.alche@eez.csic.es (J.d.D.A.); 4Department of Computer Science and Programming Languages, Universidad de Málaga, 29071 Malaga, Spain; franveredas@uma.es; 5University Institute of Research on Olive Grove and Olive Oils (INUO), Universidad de Jaén, 23071 Jaen, Spain

**Keywords:** autophagy, bioinformatics, climate change, cultivar, machine learning, multi-omics, olive, post-translational modification, programmed cell death, salinisation, salt stress, salt tolerance

## Abstract

Soil salinisation must be tackled to mitigate economic losses and challenges to global food supply caused by climate change. Olive tree, an economically relevant crop, is affected by salinisation in the Mediterranean basin. This review outlines the signs of salt stress in olive tree, as well as its morphological, physiological, and biochemical responses, together with high-throughput transcriptomics and metagenomics results obtained from salt-sensitive and -tolerant cultivars. A comprehensive list of 98 olive tree cultivars classified by salt tolerance, the list of available olive tree genomes, as well as the genes involved in salt response are provided. A successful salt-tolerant response requires at least cell wall thickening, ion exclusion, and antioxidant adaptations, likely including post-translational modifications in proteins. The promising use of soil amendments, salt-tolerant microbiota, tentative engineering of metacaspases, and integrative multi-omics tools is discussed. Olive omics’ are in its infancy, but the implementation of salt-resilient oliviculture practices and proof-of-concept trials for ‘smart oliviculture’ are making progress.

## 1. Soil Salinisation Is a Major Problem in the Context of Climate Change

### 1.1. Causes and Economical Impact of Soil Salinisation

High salt levels in soil and irrigation water are a key detrimental abiotic factor for agriculture [1,2], directly impacting food quality and quantity, as well as environmental sustainability [3]. As a result, it threatens the increase in food production required to feed rising populations, as predicted by the FAO (https://www.fao.org/soils-portal/soil-management/management-of-some-problem-soils/salt-affected-soils/more-information-on-salt-affected-soils/en/; accessed on 9 September 2024). Salinisation is a natural process (also called primary salinisation), mainly due to insufficient leaching, aeolian processes (dry deposition of oceanic salt), and physical or chemical weathering of parent rock materials (including the displacement of salt in soils to streamflow or shallow underground waters) [2]. Nowadays, however, secondary (anthropogenic) salinisation is the most concerning and devastating process, especially in the Mediterranean basin, which is an arid/semi-arid region receiving an inadequate amount of rainfall with a high evapotranspiration rate [1,2,3]. As stated by the sixth report of the United Nations’ Intergovernmental Panel on Climate Change [4], anthropogenic salinisation is caused by deforestation, irrigation with brackish or saline water, septic tank leachates, over-exploitation of groundwater (provoking a decrease in water tables and surface or subsurface seawater intrusion into coastal aquifers), and overuse of fertilisers (Figure 1, strip ‘Causes’) [1,2]. Anthropogenically induced climate change is enhancing other salinisation factors, such as an increase in Sahara dust particles lifted by the wind [5] and a decrease in precipitation and high evaporation due to global warming [2,6,7,8,9,10,11].

Temperature increases, drought waves, and the expansion of drylands are especially concerning in the Mediterranean basin, where groundwater over-exploitation is exacerbating the problem of soil salinisation [12]. Predictions suggest that soil salinisation will affect 50% of total agricultural land by 2050, resulting in significant economic losses estimated at USD 27 billion per year [13]. Currently, 33% of irrigated agricultural land and 20% of cultivated land are highly saline (>40 mM), with salinity levels expected to increase by 10% annually. This highlights the need to combat salt accumulation in the root zone [14] or to develop salt-tolerant crops [11]. Some agronomical practices may mitigate salt stress, such as restoration of vegetation cover, changes in irrigation pattern and water quality, the appropriate use of fertilisers, the application of organic amendments to soil (including microbiota changes, compost, charcoal, and chemical compounds), and the cultivation of salt-tolerant crop plants [11]. The alternative use of wastewater, while providing nutrients that may increase soil fertility, is causing unanticipated and serious salt stress [15]. In conclusion, urgent solutions are required to mitigate the effects of soil salinisation.

### 1.2. Salt Stress Is Manifested in Long-Term Exposure

The increased importance of salt stress on crops is reflected in the numerous reviews published annually on the topic ([16,17,18,19], to cite a few). Other reviews include biotechnological strategies to cope with salinity under climate change [20] and the development of salt-resilient crops [21]. Figure 1 summarises the most relevant knowledge from the causes to the signalling mechanisms that lead to adaptive and non-adaptive responses. The accumulation of water-soluble salt in the root zone causes osmotic changes that reduce the ability of plant root cells to absorb water from the soil [16,22,23]. This short-term osmotic response (Figure 1, strip ‘Short-term, primary stress’) is then followed by a long-term response [24,25] dominated by genuine salt stress adaptations (Figure 1, strip ‘Secondary stresses and adaptations’). These adaptations result in a diminished net photosynthesis rate, which in turn reduces plant growth, development, and yield [25]. In olive tree and other plants, a decrease in water potential and plant turgor was observed, leading to ABA (abscisic acid)-dependent stomata closure and a decline in photosynthesis [26]. Figure 1 includes ABA and ethylene because both phytohormones play a key role in regulating the response to salt stress [27,28]. The ABA biosynthesis is increased in roots under salt stress, initiating osmotic signalling and promoting stomatal closure to reduce water loss [29]. Ethylene is involved in signalling pathways that control cytoskeleton dynamics via microtubule repolymerisation, root development, ion transport, and senescence in response to salt stress. Another key signal is the cytosolic Ca^2+^ concentration, which increases rapidly in plants exposed to high salt concentrations. This suggests that elevated cytosolic Ca^2+^ may function as a second messenger in salt stress signalling [29,30] (Figure 1, strip ‘Signals’). Ca^2+^ signalling also facilitates the integration of the information and a physiological coordination in response to a variety of extracellular cues, such as nutrient deprivation, abiotic stresses, and intracellular signals (e.g., plant cell death [31,32]), which is vital for downstream responses. Interplay among Ca^2+^, reactive oxygen species (ROS), post-translational modifications (PTMs), phytohormones, and mitogen-activated protein kinases (MAPKs) seems to shape the acclimation responses in most plants [33].

As indicated in Figure 1 (red lines and boxes), inadequate adaptation of plants to high ion strength may lead to premature senescence and programmed cell death [34]. First signs of excessive salt exposure are chlorosis, membrane electrolyte leakage, photosynthesis impairment, reduced leaf surface and stem diameter, increased leaf thickness, compartmentalization of salt in old leaves, and stunted growth [35,36,37]. Other morphological changes include decreased stomatal density and increased trichome density [22]. Severe signs involve leaf burn, scorching, necrosis, and premature defoliation [37].

### 1.3. Why Olive Tree?

The effect of soil salinisation on many crop plants (rice, wheat, maize, sorghum, chickpea, soybean, and cotton), as well as on the model plant *Arabidopsis*, has been widely studied [38]. However, the evergreen sclerophyll olive tree (*Olea europaea* L. subsp. *europaea*), typically cultivated in the drylands of Europe and Africa, has not received the same level of attention even though it can cope with water and salt stresses [39]. This is despite the fact that the global area dedicated to olive tree cultivation spans approximately 11,594,987 ha [40], with the majority (~98%) of production coming from the salt-threatened Mediterranean basin [41], where many orchards are already suffering from soil salinisation [40,42]. Olive groves have shaped the Mediterranean landscape and have significant economic, agronomic, and agroecological implications due to their production of olive oil and table olives, which are highly valued worldwide for their health benefits [43]. In fact, these benefits are believed to be the main driver behind olive tree geographical expansion [44]. This explains why the size of olives, their oil content, and oil quality have been the focus of extensive research, which has shown that these factors depend on a complex interaction among genotype, environmental conditions, and agronomical practices [45].

Soil salinisation, which is more detrimental for olive tree when Na_2_SO_4_ is the dominant salt [26], is accepted to affect the fruit yield with a decrease in fruit weight and an increase in moisture content, without a significant alteration of oil content [40,42,46,47]. However, some existing data are somewhat contradictory. For example, there are reports indicating that olive productivity is reduced by 10% when the electrical conductivity of the soil arises 4–6 dS/m, while others report no salt stress for 90 days when NaCl is below 80 mM (7.9 dS/m) [48,49]. In other cases, a 20% increase in fruit yield was observed when salinity was increased from 4.2 to 7.9 dS/m while no significant differences were detected in terms of the basic quality parameters of olive oil, except an increase in antioxidant components [50]. Finally, other reports indicate that fruit ripening and yield is accelerated by salt stress [39], which concomitantly inhibits the linoleic desaturase activity, driving an increase in the phenol and saturated fatty acid contents and a significant decrease in the unsaturated/saturated fatty acid ratio, including the oleic/linoleic acid ratio [26,51].

Salt stress in olive groves is being exacerbated by the shift from traditional, extensive, rainfed cultivation to intensive or super-intensive groves with high water demands. Currently, 68% of olive production is rainfed, while the remaining 32% relies on irrigation and consumes in Spain about 25% of fresh water (ESYRCE 2024, by the Spanish Ministry of Agriculture, Fisheries and Food; www.mapa.gob.es/es/estadistica/temas/estadisticas-agrarias/agricultura/esyrce and https://www.mapa.gob.es/es/estadistica/temas/estadisticas-agrarias/comentariosespana2024_tcm30-697375.pdf; accessed on 24 September 2024). Interestingly, olive fruit yield under drought stress can be improved by irrigating with moderately saline water [39,52]. As long as the soil electrical conductivity in the root zone remains below 6 dS/m, drip irrigation with moderate saline water reduces plant growth and yield compared to optimal irrigation, but it still increases production compared to drought conditions [53] and the oils produced were of excellent quality [39]. In a study using a salt-tolerant cultivar, eight years of saline irrigation supplemented with Ca^2+^ showed no significant impact on growth, yield, or fruit size, and no significant accumulation of salt in the upper 30 cm of the soil (where most of the roots are located), provided that leaching by rainfall occurred at the end of the irrigation period. This suggests that saline irrigation in salt-tolerant olive groves can produce profitable agronomic results even under drought stress [54].

Sustained salt stress may result in a non-adaptive response in olive tree, which is characterised by a reduction in stalk length, leaf surface, dry weight, and root length (Figure 2), followed by irreversible leaf tip burn, leaf chlorosis, leaf rolling, wilting of flowers, and even root necrosis, shoot dieback, and defoliation [46]. Identifying and understanding the adaptive response to salt plays a pivotal role in enhancing olive grove management practices, ultimately contributing to increased yield, reduced cost, and sustainability [55]. Hence, soil amendments and salt-tolerant crops should be considered realistic short-term approaches to mitigate salt stress in olive groves. This is why this review focuses on the most recent molecular and genomic studies (in their broadest sense) performed on olive trees to gain insights into their adaptive response to salt stress, the putative roles of microbiota, and future approaches based on artificial intelligence, with the aim of maintaining olive yields in stressed lands.

## 2. Salt Tolerance in Olive Tree Is Cultivar-Dependent

The olive tree has been described as an intermediate tolerant species to salinity when compared to other fruit trees. The salt tolerance threshold was established at 8 mg/mL (137 mM or 13.4 dS/m) NaCl [48,49]. Besides the cultivar-dependent level of tolerance [48], the natural habitat of olive tree is calcareous soil, where the available Ca^2+^ mitigates the toxic effects of Na^+^ on plasma membrane integrity and prevents the transport of Na^+^ and Cl^−^ to sensitive shoots and leaves [56]. This may explain, at least partially, why olive tree is more tolerant to salt than other fruit trees. Furthermore, olive groves featuring salt-tolerant trees may have positive effects on the soil, as salt-sensitive trees struggle to capture nitrogen compounds in salinised soils, while salt-tolerant trees are better able to do so [57]. Hence, salt-tolerant cultivars or rootstocks are promising for cultivating salinised lands.

As with other fruit trees, where rootstocks have a great influence on plant vigour, nutritional status, and response to biotic and abiotic stresses [58,59], olive tree behaviour is also affected by the rootstock used [60]. For example, grafting disease-susceptible olive scions onto disease-resistant olive rootstocks can reduce their susceptibility to *Verticillium* wilt [61]. Grafting on wild tree rootstocks is a traditional method for producing stronger trees with improved fruit quality, but, unfortunately, it has been increasingly replaced by modern self-rooting techniques [62]. Wild olive trees exhibit high genetic variability, and therefore are a valuable source of rootstocks with genes resistant to abiotic stresses [62]. Furthermore, salt-tolerant rootstocks can reduce the toxic effects of salinity in grapevines [63]. Hence, using salt-tolerant rootstocks bearing salt-sensitive scions is expected to mitigate salt stress in olive tree. The use of salt-tolerant cultivars or rootstocks should be encouraged in salinised soils in the short and medium term to avoid the time-consuming process of breeding salt-tolerant plants [64].

Classifying olive cultivars and rootstocks based on their salt tolerance levels has become a crucial task. SILVOLIVE is an important source for characterised rootstocks [62], but no similar database exists for olive cultivars. As a result, information about the salt tolerance of olive cultivars was compiled from the literature and is presented in Table 1, Table 2, Table 3 and Table 4. Unfortunately, the methods used to asses salinity tolerance are inconsistent, with some studies providing reliable results based on multiple parameters [65,66] but focusing on only a few cultivars (both articles provide inconsistent results for ‘Arbequina’), while others classify up to 26 cultivars based on a single parameter [67]. Several factors contribute to this heterogeneity, including (1) the treatment duration, which varied between studies (e.g., 30 days for ‘Kilis’ (Table 1) and ‘Gemlik’ (Table 4) compared to 9 years for ‘Barnea’ (Table 1)); (2) the strength of the salt treatment, which is difficult to compare as it was often described using different units, such as molarity (mM), milliequivalents (meq), mg/L, parts per million (ppm), or even soil electrical conductivity (measured in different Siemens units like dS/m, dS/cm, or mS/cm) and, more surprisingly, with some reports lacking any description (ND in Table 1, Table 2 and Table 3); (3) the use of NaCl as the primary though other reports use a mix of salts; and (4) the age of the plants used, which ranges from two-month old plantlets [67] to one year old plants [66]. Here, we classified 90 cultivars as salt-tolerant (31; Table 1), intermediate tolerant (33; Table 2), and salt-sensitive (26; Table 3). The salt tolerance classification of eight cultivars was consistent in different reports (‘Chemlali’, ‘Frantoio’, ‘Kalamon’, ‘Megaritiki’, and ‘Royal de Cazorla’ in Table 1, ‘Manzanillo’ in Table 2, and ‘Chondrolia Chalkidikis’ and ‘Leccino’ in Table 3), but another eight cultivars were unclassifiable due to inconsistent findings (Table 4).

Concerning the salt-tolerant cultivars (Table 1), ‘Frantoio’ is commonly considered in this group. However, in one study [68], ‘Frantoio’ was described as less salt-tolerant than ‘Ocal’ (Table 1) and ‘Picudo’ (Table 4), which led authors to classify it as intermediate.

**Table 1 biology-14-00287-t001:** Salt-tolerant olive cultivars based on analytical data or common usage.

Cultivar	Reference	Country	Salt in Irrigation Water	Treatment Time
Abou-Satl	[66]	Syria	12 dS/m NaCl	90 days
Amigdalilolia	[69]	Iran	12 dS/m NaCl	90 days
Ayvalık	[70]	Turkey	300 mM NaCl	30 days
Barnea	[49]	Israel	7.5 dS/m	9 years
Cañivano	[67]	Spain	100 mM NaCl	49 days
Chemlali	[49]	Tunisia	brackish water	8 months
	[71]		200 mM NaCl	5 months
Dakal	[69]	Iran	12 dS/m NaCl	90 days
Empeltre	[72]	Spain	10 dS/m	4 years
Escarabajuelo	[67]	Spain	100 mM NaCl	49 days
Frantoio	[73]	Italy	200 mM NaCl	60 days
	[74]		100 mol/m^3^ SIW ^a^	ND
	[68]		200 mM NaCl	84 days
	[75]		200 mM NaCl	56 days
Hamed	[49]	Egypt	6000 ppm NaCl, Na_2_SO_4_, CaCl_2_, and MgSO_4_	2 years
Istarska bjelica	[76]	Croatia	100 mM NaCl	70 days
Jabaluna	[67]	Spain	100 mM NaCl	49 days
Jlot	[77]	Syria	8000 ppm NaCl	ND
Kalamata	[49]	Greece	200 mM NaCl	5 months
Kalamon	[78]	Greece	120 mM NaCl	90 days
	[79]		120 mM NaCl	90 days
Kerkiras	[49]	Greece	ND	ND
Kilis	[70]	Turkey	300 mM NaCl	30 days
Kilis Yağlık	[80]	Turkey	150 mM NaCl	4 months
Koroneiki I38	[65]	Spain	200 mM NaCl	5 months
Kothreiki	[49]	Greece	200 mM NaCl	5 months
Lechín de Sevilla	[67]	Spain	100 mM NaCl	49 days
Lianolia	[49]	Greece	ND	ND
Megaritiki	[73]	Greece	150 meq NaCl	4 months
	[81]		200 mM NaCl	5 months
Nevadillo	[67]	Spain	100 mM NaCl	49 days
Oblica	[76]	Croatia	100 mM NaCl	70 days
Ocal	[68]	Spain	200 mM NaCl	84 days
Pocama	[82]	Egypt	6000 ppm NaCl, Na_2_SO_4_, CaCl_2_, and MgSO_4_	2 years
Royal de Cazorla	[83]	Spain	100 mM and 200 mM NaCl	8 months
	[84]		100 mM and 200 mM NaCl	60 days
	[85]		200 mM NaCl	8 months
Verdale	[82]	France	6000 ppm NaCl, Na_2_SO_4_, CaCl_2_, and MgSO_4_	2 years
‘Zeitoun Ennour’ ^b^	[64]	Tunisia	225 mM (21.8 mS/cm) NaCl	6 months

^a^ SIW: saline irrigation water prepared by mixing chloride, sulphate, and bicarbonate of Na, Ca, and Mg. ^b^ New cultivar resulting from genetic breeding. ND: not described.

The middle level of salt tolerance was qualified as ‘intermediate’ (Table 2). This group includes ‘Changlot Real’ [67], but information from nurseries (https://gardencenterejea.com/olivos/382-olivo-changlot-realacebuche-c1.html; accessed on 17 October 2024) and the experience of the present authors (Figure 2) indicate that it can tolerate high salt levels and should be preferably qualified as a salt-tolerant cultivar.

**Table 2 biology-14-00287-t002:** Olive cultivars with intermediate tolerance to salt stress based on analytical data or common usage.

Cultivar	Reference	Country	Salt in Irrigation Water	Treatment Time
Adramitini	[49]	Greece	ND	ND
Aggezi	[49]	Egypt	6000 ppm NaCl, Na_2_SO_4_, CaCl_2_, and MgSO_4_	2 years
Alameño	[67]	Spain	100 mM NaCl	49 days
Arbosana I43	[71]	Spain	200 mM NaCl	5 months
Blanqueta	[67]	Spain	100 mM NaCl	49 days
Cañivano Negro	[67]	Spain	100 mM NaCl	49 days
Carolea	[73]	Italy	14.97 mS/cm	11 months
Casta Cabra	[68]	Spain	200 mM NaCl	84 days
Changlot Real	[67]	Spain	100 mM NaCl	49 days
Chorruo	[67]	Spain	100 mM NaCl	49 days
Coratina	[73]	Italy	200 mM NaCl	60 days
Cornicabra	[68]	Spain	200 mM NaCl	84 days
Dezful	[69]	Iran	12 dS/m NaCl	90 days
Gordal Sevillana	[67]	Spain	100 mM NaCl	49 days
Hojiblanca	[67]	Spain	100 mM NaCl	49 days
Khaisi	[77]	Syria	8000 ppm NaCl	ND
Lechín de Granada	[67]	Spain	100 mM NaCl	49 days
Manzanilla de Sevilla	[67]	Spain	100 mM NaCl	49 days
Manzanillo	[73]	Spain	60 mM NaCl	115 days
	[86]		4000 mg/L MgSO_4_, CaSO_4_, NaCl, MgCl_2_, and CaCO_3_	9 months
	[82]		6000 ppm NaCl, Na_2_SO_4_, CaCl_2_, and MgSO_4_	2 years
Maraiolo	[49]	Italy	ND	ND
Maurino	[49]	Italy	200 mM NaCl	60 days
Moraiolo	[73]	Italy	200 mM NaCl	60 days
Mostazal	[49]	Peru	6000 ppm NaCl, Na_2_SO_4_, CaCl_2_, and MgSO_4_	2 years
Nabali Muhassan	[49]	Jordan	100 mol/m^3^ SIW ^a^	ND
Nizip Yağlik	[80]	Turkey	150 mM NaCl	4 months
Oblonga	[67]	Italy	100 mM NaCl	49 days
Redondil	[67]	Spain	100 mM NaCl	49 days
Toffahi	[49]	Egypt	6000 ppm NaCl, Na_2_SO_4_, CaCl_2_, and MgSO_4_	2 years
Tokhm-e-Kabki	[69]	Iran	12 dS/m NaCl	90 days
Valanolia	[49]	Greece	ND	ND
Verdial de Vélez-Málaga	[67]	Spain	100 mM NaCl	49 days
Wardan	[49]	Egypt	6000 ppm NaCl, Na_2_SO_4_, CaCl_2_, and MgSO_4_	2 years
Zorzariega	[67]	Spain	100 mM NaCl	49 days

^a^ SIW: saline irrigation water prepared by mixing chloride, sulphate, and bicarbonate of Na, Ca, and Mg. ND: not described.

The salt-sensitive cultivars (Table 3) are unsuitable for salinised soils and are not recommended for planting in those regions, unless they are grafted on a salt-tolerant rootstock.

**Table 3 biology-14-00287-t003:** Salt-sensitive olive cultivars based on analytical data or common usage.

Cultivar	Reference	Country	Salt in Irrigation Water	Treatment Time
Aggezi Shami	[86]	Egypt	4000 mg/L MgSO_4_, CaSO_4_, NaCl, MgCl_2_, and CaCO_3_	275 days
Aguromanaki	[49]	Greece	ND	ND
Arbequina I18	[71]	Spain	200 mM NaCl	5 months
Arbosana	[65]	Spain	200 mM NaCl	5 months
Bezeri	[77]	Syria	8000 ppm NaCl	ND
Bouteillan	[49]	France	6000 ppm NaCl, Na_2_SO_4_, CaCl_2_, and MgSO_4_	2 years
Buža	[76]	Croatia	100 mM NaCl	70 days
Chondrolia Chalkidikis	[78]	Greece	120 mM NaCl	90 days
	[87]		150 meq NaCl	4 months
	[78]		120 mM NaCl	90 days
	[88]		75 mM NaCl	45 days
	[89]		75 mM and 150 mM NaCl	45 days
Cobrançosa	[67]	Portugal	100 mM NaCl	49 days
Conservalia	[69]	Iran	12 dS/m NaCl	90 days
Drobnica	[76]	Croatia	100 mM NaCl	70 days
Fadaq86	[85]	Iran	200 mM NaCl	8 months
Galego	[67]	Spain	100 mM NaCl	49 days
Khederi	[77]	Syria	8000 ppm NaCl	ND
Lastovka	[76]	Croatia	100 mM NaCl	70 days
Leccino	[73]	Italy	200 mM NaCl	60 days
	[76]		100 mM NaCl	70 days
	[75]		200 mM NaCl	56 days
Mastoidis	[90]	Greece	200 mM NaCl	5 months
Meski	[67]	Tunisia	100 mM NaCl	49 days
Mission	[82]	US	6000 ppm NaCl, Na_2_SO_4_, CaCl_2_, and MgSO_4_	2 years
Nabal	[49]	Egypt	6000 ppm NaCl, Na_2_SO_4_, CaCl_2_, and MgSO_4_	2 years
Nocellara del Belice	[91]	Italy	14.97 mS/cm	11 months
Pajarero	[73]	Spain	100 mM NaCl	49 days
Rosciola	[82]	Italy	6000 ppm NaCl, Na_2_SO_4_, CaCl_2_, and MgSO_4_	2 years
Shiraz	[69]	Iran	12 dS/m NaCl	90 days
Sourani	[77]	Syria	8000 ppm NaCl	ND
Throubolia	[49]	Greece	ND	ND

ND: not described.

The last eight cultivars (including the widely used ‘Arbequina’, ‘Koronieki’, and ‘Picual’) were assigned to different levels of salt tolerance depending on the published studies (Table 4). The most erratic cases correspond to ‘Arbequina’ and ‘Picual’ (qualified as tolerant, intermediate, and sensitive) followed by ‘Chétoui’ and ‘Gemlik’ (qualified as sensitive or tolerant). Classification by relative comparison, divergences in salt concentration and treatment length, and the age of treated plants might partially explain the inconsistencies in tolerance qualification. This situation also supports the classic idea that techniques to measure salt tolerance require more standardisation, perhaps excluding fruit yield or growth-related parameters [26,67,92].

**Table 4 biology-14-00287-t004:** Olive cultivars with inconsistent literature classification according to their salt tolerance.

Cultivar	Tolerant	Intermediate	Sensitive	Country	Salt in Irrigation Water	Treatment Time
Amphissis		[87]		Greece	150 meq NaCl	4 months
			[90]		200 mM NaCl	5 months
Arbequina	[73]			Spain	100 mM NaCl	49 days
	[72]				10 dS/m	4 years
		[65]			200 mM NaCl	5 months
		[84]			100 mM and 200 mM NaCl	60 days
			[66]		12 dS/m NaCl	90 days
Chétoui	[71]			Tunisia	200 mM NaCl	5 months
			[67]		100 mM NaCl	49 days
Gemlik	[70]			Turkey	300 mM NaCl	30 days
			[80]		150 mM NaCl	4 months
Koroneiki		[82]		Greece	6000 ppm NaCl, Na_2_SO_4_, CaCl_2_, and MgSO_4_	2 years
		[87]			150 meq NaCl	4 months
			[84]		100 mM and 200 mM NaCl	60 days
			[83]		100 mM and 200 mM NaCl	8 months
			[85]		200 mM NaCl	8 months
			[71]		200 mM NaCl	5 months
			[90]		200 mM NaCl	5 months
Picual	[73]			Spain	100 mM NaCl	49 days
	[86]				4000 mg/L MgSO_4_, CaSO_4_, NaCl, MgCl_2_, and CaCO_3_	9 months
	[82]				6000 ppm NaCl, Na_2_SO_4_, CaCl_2_, and MgSO_4_	2 years
		[84]			100 mM and 200 mM NaCl	60 days
			[85]		200 mM NaCl	8 months
			[68]		200 mM NaCl	84 days
Picudo	[68]			Spain	200 mM NaCl	84 days
		[67]			100 mM NaCl	49 days
Zard	[66]			Iran	12 dS/m NaCl	90 days
		[69]			12 dS/m NaCl	90 days

## 3. Mechanisms of Salt Adaptation in Olive Tree

Salt tolerance is a polygenic trait influenced by both genetic and environmental factors [93], enabling olive trees to cope with the different stresses and dysregulations induced by the excess of salts in the root zone [94].

### 3.1. Root Cell Wall Thickening

The epidermis is the first line of defence against environmental stresses in many plants, including olive tree [95]. In the case of salt stress, roots play a critical role in signalling and managing the excess of salt. Roots are highly vulnerable to salt [96], yet they adapt by increasing the root-to-shoot ratio to diminish the demand for element supply, and to improve water absorption, and to maintain the osmotic balance [97]. Olive tree roots exposed to high salt concentrations modify their structure to enhance salt compartmentation as follows: reducing root number and length (Figure 2), increasing root turnover and diameter, and raising the root concentration of Na^+^ and Cl^−^ by 5 to 10 times with respect to leaf tissues. Clearly, protecting aerial tissues from salt imposes a high cost on the roots [95,98]. These modifications may be mediated by cytoskeletal rearrangements regulated by Ca^2+^ levels, the assembly of the cellulose synthase (CesA) complexes (CSCs), and the cellulose microfibril organisation [99]. Consequently, the synthesis and deposition of epicuticular wax and other cell wall components (such as cellulose, pectins, hemicelluloses, lignin, suberin, and waxes) are enhanced to prevent water loss, accompanied by a decrease in the transport of ion surplus to the xylem. This response is regulated by several *MYB* transcription factors, as well as members of the *WRKY*, *ERF*, and *NAC* gene families [100]. Ultimately, this leads to an increase in both cell wall and leaf thickness mainly through suberisation and lignification (Figure 1, strip ‘Long-term, adaptive response’) [101,102]. These events were reported in halophyte and glycophyte roots, including olive tree, as a response to salt stress, helping to prevent transport of external ions to the xylem stream [103,104,105]. Hence, cell wall thickening in olive roots is a dominant trait triggered by salt stress in both salt-tolerant and -sensitive cultivars, although this alone does not confer salt tolerance in salt-sensitive cultivars [104].

### 3.2. Osmoprotection

Salt stress induces an early osmotic stress (Figure 1, strip ‘Secondary stresses and adaptations’) [28]. As a result, most crop plants synthesise and accumulate osmolytes or osmoprotectants (sugars, proline, arginine, glycine betaine, organic acids, etc.) [106] to maintain cellular osmotic potential, turgor, and protein stability. In cultivars like ‘Picual’, ‘Gemlik’, and ‘Chemlali’, proline accumulates in roots and leaves in response to salt due to its high hydration capacity. Proline binds to proteins, preventing their denaturation [68,107,108,109]. However, in cultivars ‘Chétoui’, ‘Arbequina’, ‘Koroneiki’, ‘Royal de Cazorla’, and ‘Fadak 86’, proline levels decrease under salt stress [83,103]. Other osmoprotective responses in olive tree to counteract oxidative damage under salt stress include the accumulation of mannitol, violaxanthin-cycle pigment, and flavonoids [110].

### 3.3. Antioxidant Adaptations

Salt stress also induces oxidative stress (Figure 1, strip ‘Secondary stresses and adaptations’), characterised by the overproduction of reactive oxygen species (ROS) [28]. Under normal conditions, ROS act as secondary messengers for growth and development, and can be managed by the common antioxidant defences [33,94]. The abnormal production of ROS due to salt exposure can overwhelm these defence mechanisms [94], resulting in oxidative damage of macromolecules and the disorganisation of cell membranes, which worsens the effects of salinity [24,111]. This damage can lead to ion leakage, histone modifications, DNA methylation, and alternative splicing [16,112], as well as changes in membrane and protein surface charges [112]. The complex antioxidant defence system consists of enzymes [113] (e.g., superoxide dismutase [SOD], catalase, peroxidases, glutathione reductase, and glutathione-*S*-transferase) and non-enzymatic components (e.g., ascorbic acid, tocopherol, carotenoids, phenolic compounds, alkaloids, and glutathione) [24,114]. In olive tree leaves, ROS levels rise under salt stress, prompting an increase in cultivar-dependent antioxidant activity [83]. The degree of salt tolerance in olive tree cultivars can be inferred from the time required to increase SOD levels, as more sensitive genotypes accumulate higher concentrations of SOD more quickly [115].

Olive tree also accumulates phenolic compounds, mainly oleuropein, as part of its non-enzymatic antioxidant defence. A significant correlation was observed between total phenol content and antioxidant activity in both leaves and roots across different olive cultivars [116]. The salt-sensitive ‘Leccino’ shows a strong upregulation of key genes in the phenylpropanoid pathway under salt stress, while the salt-tolerant ‘Frantoio’ shows less intense upregulation of those pathways but a higher phenolic compound content. Thus, ‘Frantoio’ combines cell wall thickening [104], antioxidant defence activation [75,117], and Na^+^ exclusion (see below) to address NaCl toxicity.

### 3.4. Ion Detoxification

Salt stress induces ionic stress (Figure 1, strip ‘Secondary stresses and adaptations’) by disturbing ionic homeostasis [28]. Halophyte plants, adapted to saline conditions of 70–200 mM NaCl [118], contrast with glycophytes, which require soils with a low salt content to maintain low Na^+^ and Cl^–^ levels in their aboveground tissues [119]. Na^+^ in the cytosol is detrimental in both glycophytes and halophytes [16,25,114], including many crops [25,36,120,121], making ion exclusion crucial to prevent its intracellular accumulation in aerial tissues. The chemical similarity between Na^+^ and K^+^ causes high levels of external Na^+^ to competitively inhibit K^+^ uptake systems and stimulate K^+^ efflux, leading to insufficient intracellular K^+^ concentrations [122,123,124]. Maintaining balanced cytosolic Na^+^/K^+^ ratios becomes thus a key tolerance mechanism. Transporters likely prevent excessive Na^+^ accumulation and minimise K^+^ loss in the cytosol to maintain a high K^+^/Na^+^ ratio [16,28]. Key ion transporters include (1) NSCC (Non-Selective Cation Channel), AKT1 (Arabidopsis K^+^ Transporter 1), HAK5 (High Affinity K^+^ transporter 5), and SKOR (Stelar K^+^ Outward Rectifying channel)/GORK (Guard Cell Outward Rectifying K^+^ channel), and plasma membrane H^+^-ATPases, involved in selective K^+^ vs. Na^+^ uptake in roots; (2) Na^+^/H^+^ antiporters, SOS1 (Salt Overly Sensitive 1), and plasma membrane ATPases for Na^+^ removal to the apoplast; (3) Na^+^/H^+^ antiporters, NHXs (Na^+^/H^+^ Exchangers), vacuolar H^+^-ATPase, and vacuolar H^+^-PPase (pyrophosphatase), involved in vacuolar and endosomal compartmentalisation; and (4) HKT1 (High-Affinity K^+^ transporter 1) and SOS1, involved in xylem and phloem loading and unloading. A recent study proposed that some glycophytic plants, mainly crops, have evolved mechanisms that allow them to tolerate increased Na^+^ accumulation in leaves by a yet unknown mechanism likely related to HKT variants [125].

Excessive Cl^–^ accumulation (but not Na^+^) in leaves correlates with a decrease in transpiration, photosynthesis, yield, and quality, eventually leading to plant death in many fruit trees [22,126,127,128,129]. However, mechanisms preventing Cl^–^ toxicity have been poorly studied despite the identification of several genes exhibiting Cl^–^ transport activity, such as *NPF* (Nitrate transporter 1/Peptide transporter Family), *NRT2* (NitRate transporter 2), *SLAH* (SLow-type Anion channel Homologs), *SLAC* (SLow Anion Channel-associated), *CCC* (Cation-Cl^–^ Cotransporter), *ICln* (Ionic Conductance L-type Nonspecific), *ALMT* (ALuminum-activated Malate Transporter), *CLC* (ChLoride Channel), or *NAXT* (Na^+^/H^+^ eXchanger Transporter) [128,130,131]). In a recent study, a new mechanism of Cl^–^ exclusion was proposed in which the chloride channel protein *f* (CLCf) in *Arabidopsis thaliana* moves from Golgi’s apparatus to the plasma membrane to remove the excess of Cl^–^ when plants are treated with NaCl [132]. This change in location resembles the behaviour of VPEs (vacuolar processing enzymes [133]), which are involved in the execution of programmed cell death (PCD; see below).

Other ion-detoxifying mechanisms involve (1) aquaporins from the PIP (plasma membrane intrinsic proteins) family, which accumulate in the root cortex to reduce the water conductance [134]; (2) dehydrins, which increase their expression to prevent dehydration-induced cellular damage [134]; and (3) tetraspanins (integral membrane proteins characterised by four transmembrane domains), which maintain an elevated K^+^/Na^+^ ratio to gain ionic equilibrium, promote proline accumulation, and regulate antioxidant enzymes [135].

The level of tolerance to salt in olive tree is closely associated with effective mechanisms of Na^+^ and Cl^−^ exclusion in leaves and retention in roots, preventing ion accumulation in actively growing shoots [71,81,90]. Salt-sensitive shoots are also protected though moderate growth and reduced water transport, preventing salt from reaching aerial tissues [92]. Ions accumulate in the stem and old leaves only after prolonged exposure to salinity (above 6–8 dS/m) [26,56]. Ion exclusion, driven by a drastic decline in water mass flow, seems an untenable strategy for most annual crops, but as olive cultivars have evolved under adverse environments, they are able to assimilate CO_2_ and produce new growth at a considerable rate, allowing for rapid recovery when good-quality water becomes available [110]. Hence, Na^+^ and Cl^−^ concentrations in leaves are good indicators of salt stress [71,98] with damage occurring when Na^+^ and Cl^–^ levels exceed 0.5% and 0.2% dry weight, respectively [46]. Stem elongation and gas exchange in olive leaves may be misleading, as they are not strongly correlated with Na^+^ and Cl^–^ levels or salt treatments in olive tree [26,92]. Despite a decrease in net photosynthesis, leaf chlorophyll content, and leaf dry weight [83], there is no relationship between salt accumulation and photosynthesis decline in either young or old leaves [136]. This indicates that the photosynthetic sensitivity of olive tree to salt does not depend on salt exclusion or compartmentalisation in old leaves [136]. Growth parameters such as shoot elongation, trunk diameter, plant dry weight, number of leaves, leaf thickness, internode length, total leaf area, and shoot/root ratio are reliable surrogates for assessing salt effects on olive tree [40].

‘Frantoio’ shows a higher mobilisation of Ca^2+^ and increased expression of transporter activities in the cytoplasmic membrane rather than the tonoplast [137]. This mechanism may be similar to the recently described tolerance mechanism involving CLCf in *A. thaliana* [132]. A comparative study [138] revealed that salinised olive trees (100 mM NaCl) had a lower osmotic potential compared with citrus trees (50 mM NaCl), with Cl^–^ and Na^+^ concentrations increasing in the leaves and roots of both species, while a decrease in leaf chlorophyll *a* was only observed in citrus. As a result, young olive plantlets appear to show no apparent damage under salinity (6.6 dS/m) [46], while adult trees under moderate salinity (40–60 mM NaCl) experience a decrease in pollen viability and germinability, a lower number of perfect flowers per inflorescence, and decreased fruit set [46]. High salt treatments also lead to leaf drop [71].

### 3.5. Salt-Adaptive Responses of Emerging Interest

There are other emerging aspects of salt stress responses that should be addressed to have a deeper knowledge of olive tree management of salt stress as well as how to use this knowledge to improve plant salt tolerance. Besides the miRNAs that contribute to salt tolerance, whose importance in olive tree has not been addressed yet [139], other interesting studies are related to the PTM of proteins, programmed cell death, and autophagy.

#### 3.5.1. Post-Translational Modifications

Recent research has demonstrated that protein PTMs are a key adaptive signalling mechanism in plants. While the role of PTMs in plant salt tolerance is still in its early stages, research has demonstrated the importance of ubiquitination and SUMOylation in regulating salt stress, as well as the importance of phosphorylation in ion homeostasis under salt stress [140]. PTMs provide salt tolerance by modulating the activity of numerous enzymes and proteins (including transporters), modifying their cell localisation or binding capacity, and minimising ROS production [140]. Additionally, salt stress triggers ROS, reactive nitrogen species (RNS), and other free radicals like reactive sulphur species (RSS), which exert their salt-tolerant signalling functions through other PTMs, such as carbonylation, *S*-nitrosation and Tyr-nitration, sulfhydration, and persulphidation [141,142]. Fatty acid-dependent PTMs (e.g., *S*-acylation), *N*-acetylation, and *N*-glycosylation are also promising PTMs for enhancing salt tolerance [140].

Various chemicals known to induce PTMs, such as hydrogen sulphide (H_2_S), NO, hydrogen peroxide (H_2_O_2_), and melatonin, were shown to enhance salt tolerance in crops when used for priming seeds and seedlings [143,144], probably by means of the PTMs they cause. PTM-primed salt-tolerant crops could be used in salinised lands without the need for genetic engineering, time-consuming breeding programmes, or replanting with salt-tolerant crops.

The molecular response to salt stress in the roots and leaves of one-year-old olive plants was investigated after salt priming in the salt-sensitive cultivar ‘Chondrolia Chalkidikis’ using a multi-omics approach [89]. Among other findings, this study revealed the key role of protein PTMs in both leaves and roots, particularly phosphorylation, carbonylation, and *S*-nitrosation. The authors suggested that these PTMs do not merely reflect epiphenomena of salt stress but rather serve as adaptive mechanisms to optimise olive tree metabolism under the oxidative/nitrosative conditions induced by salt priming. They also proposed that PTMs may be more relevant than transcriptional reprogramming in responding to salt stress in olive trees.

#### 3.5.2. Programmed Cell Death

Programmed cell death (PCD) is often overlooked in plant stress studies, despite being a key strategy evolved by plants to tolerate extreme environments by systematically removing superfluous and damaged cells. Na^+^ toxicity (rather than osmotic stress) leads to PCD activation [145] (Figure 1, red lines and boxes). The Na^+^/K^+^ imbalance caused by salt stress is crucial for triggering the effector enzymes responsible for PCD [146,147]. Salt treatments lead to a rapid rise in cytosolic Ca^2+^ levels, initiating a Ca^2+^/SOS cascade that distinguishes salt stress from osmotic stress [32], ultimately triggering PCD and activating death-specific enzymes [31]. PCD relies on both caspases and vacuolar processing enzymes (VPEs), which are typically located in the apoplast under normal conditions but are relocated to the cytoplasm under stress. VPEs were also recently identified as executors of NaCl stress-induced vacuole-mediated PCD [133]. The overexpression of anti-apoptotic proteins such as Bcl-2 inhibits salt-induced PCD by suppressing the increase in Ca^2+^ levels and decreasing VPE expression, thereby improving salt stress tolerance [148]. Hence, VPEs (central mediators of salt stress-induced PCD in plants) may enhance plant salt tolerance in a transporter-independent manner. Recent multi-omics studies on olive tree have found that KTI2 (Kunitz trypsin inhibitor 2), a protein with chlorophyll-binding capabilities involved in cell death regulation, is differentially expressed in salt primed olive plants [89].

#### 3.5.3. Autophagy

To mitigate high salt concentrations in the cytosol, autophagy—a process that helps plants to resist various environmental stresses [149]—is induced during salt stress (Figure 1, red boxes and lines). NaCl treatments induce the expression of several autophagy-related genes (ATGs) in plants, facilitating Na^+^ sequestration in the central vacuole, mediated by metacaspases [150]. Plant metacaspases not only regulate and execute autophagy–PCD crosstalk [151,152] in a Ca^2+^-dependent manner [153,154], but also increase their expression in response to salt stress [152]. Consequently, metacaspases have emerged as promising biotechnological targets for improving plant salt tolerance through the manipulation of the innate vacuolar PCD pathway.

## 4. Genomics of Salt Stress in Olive Tree

### 4.1. Genome Sequences of Olive Cultivars

In recent years, olive tree genome sequences have started to populate databases (Table 5). The first sequenced genome was that of a very old ‘Farga’ tree located in Spain [155], whose 23 chromosome assembly and annotations were recently improved [156]. Next came the genome of wild olive, or oleaster (*O. europaea* L. subsp. *europaea* var. *sylvestris*), from an individual located in Turkey [44]. It was recently suggested, however, that this olive tree is closer to feral than truly wild [156]. The genome of ‘Picual’, widely cultivated in Spain and thought to have intermediate salt tolerance (Table 4 and Figure 2), was published at the scaffold stage [157]. Another widely cultivated olive tree cultivar, ‘Arbequina’, also considered to have intermediate salt tolerance (Table 4 and Figure 2), was recently sequenced at the chromosome level, combining short- and long-reads [158]. Using the latest advances in sequencing platforms, the first gap-free, telomere-to-telomere olive tree genome was published for the salt-sensitive cultivar ‘Leccino’ (Table 1) [159]. This genome (Table 5, T2T-Lec version) represents the highest contiguous and complete olive genome to date. More recently [160], a preprint describes scaffold-level assemblies of ‘Leccino’ and ‘Frantoio’ genomes (Table 5). There are also two genome sequences of *Olea europaea* subsp. *cuspidata* (Table 5), a wild olive subspecies native to Northeast Africa and Southwest Asia, extending to drier regions of Yunnan and Sichuan in China. This subspecies, considered the ancestor of the European olive, was initially introduced into Australia and New Zealand as an ornamental plant, but it is now regarded as an important rootstock due to its fungal resistance, robust growth, and high survival rate for olive cultivar grafts [161,162,163]. Finally, sequencing projects for the Algerian *Olea europaea* subsp. *laperrinei* (GCA_045282215.1 and GCA_045282115.1) are currently ongoing.

Table 5 indicates that the number of protein-coding genes in olive tree cultivars ranges from 67,103 to 53,518, with the higher number found in ‘Leccino’ and the lowest number in ‘Arbequina’. The wild type, however, appears to have a significantly lower number of genes (47,911). Interestingly, both the wild olive and the *cuspidata* ancestors have a small yet similar number of genes, ranging from 43,511 to 46,904. Sebastiani and colleagues [160] examined structural differences between ‘Leccino’ (salt-sensitive, Table 3) and ‘Frantoio’ (salt-tolerant, Table 1) genomes, focusing on nine key genes related to salt stress. They identified 15 structural variations, among which those of *ATPase11* and *ATPase8* genes in ‘Frantoio’ present exclusive structural variations that might explain the overexpression of these genes only when this cultivar is salt stressed. In contrast, the absence of variations in Na^+^/H^+^ antiporter gene *SOS1* suggests that changes in its expression do not contribute to the salt stress tolerance of ‘Frantoio’. It is expected that the release of telomere-to-telomere assemblers like Verkko [164] will foster the assembly of gapless olive genomes, bridging the gap between olive genomics and the features influencing salt stress [165].

### 4.2. Low-Throughput Transcriptional Studies

Over the last decade, many key genes for salt tolerance have been reported in plants, most of them involved in oxidation–reduction processes, ion transport, chloride channels, cell wall synthesis, and hormone-related genes, supporting the idea that salt tolerance is a multigenic response [28]. Plant transcription factor families such as MYB, WRKY, basic leucine zipper (bZIPs), basic helix–loop–helix (bHLH), NAC, homeobox (HB), and GATA are key in the early response to salt stress [166,167]. Table 6 summarises the most relevant olive genes whose expressions change in salt stress in one cultivar, or between salt-tolerant and -sensitive cultivars.

The transcriptomic response to salinity was profiled on olive cultivars using qRT-PCR, focusing on salt tolerance genes described in other species. The salt-sensitive ‘Chondrolia Chalkidikis’ [88] genes involved in tyrosine (*PPO* and *hisC*), flavonoid (*F3H*, *FNSII*, and *CA4H*), lignan (*PLRTp2*), and secoiridoid (*GTF*) metabolism (Table 6) was investigated. The tyrosine and flavonoid metabolism-related genes were clearly modulated by salt in old leaves, while *GTF* gene upregulation in new leaves demonstrated that oleuropein metabolism was modified by salt stress. Using the salt-sensitive ‘Leccino’ [51], a significant reduction in FAD6 transcript levels, together with a synergic upregulation of the *SAD1* gene, was associated with an increase in the oleic/linoleic and a decrease in the polyunsaturated/monounsaturated ratios in the maturing fruit mesocarp.

Shoots of high- and medium-tolerance Spanish cultivars (‘Royal de Cazorla’ and ‘Picual’, respectively) were profiled and compared with low-tolerance Greek (‘Koroneiki’) and Iranian (‘Fadak86’) varieties [85]. Up to 10 salt-responsive candidate genes were tested, but only the *NHX7*, *P5CS*, *RD19A*, and *PetD* genes were upregulated in salt-tolerant cultivars (Table 6). These data suggest that the maintenance of ionic homeostasis, the increase in turgor pressure, the modulation of electron transport and the generation of ATP, and the induction of stress-related proteins (SRPs) are key processes to cope with salt stress in olive tree. Moreover, the expression of the *PIP1.1* (an aquaporin), *PetD* (a cytochrome b6), *PI4Kg4* (a phosphatidylinositol 4-kinase), and *XylA* (xylose isomerase A) genes (Table 6) are differentially regulated by DNA methylation among cultivars, suggesting that this mechanism could contribute to the adaptation to high salt of the salt-tolerant cultivar ‘Royal de Cazorla’ [84]. Finally, using ‘Picual’ (tolerant) and ‘Nabali’ (moderately tolerant) cultivars [168], novel salt-responsive genes (*MO1*, *STO*, *PMP3*, *USP2*, *AP-4*, *WRKY1*, *CCX1*, and *KT2*; Table 6) were differentially expressed, suggesting that different sets of salt-responsive genes are triggered depending both on the salt concentration and the cultivar genotype.

### 4.3. High-Throughput Transcriptomics

A pioneer study using microarrays [78] based on the ‘Kalamon’ (salt-tolerant) and ‘Chondrolia Chalkidikis’ (salt-sensitive) cultivars found the upregulation of 159 genes (and de novo synthesis of 50 genes) in roots of the salt-tolerant cultivar under salt stress. All differentially upregulated genes decrease their expression level after the plants recovered from stress. In the salt-sensitive cultivar, only 20 genes responded to salt stress. When the same authors used 454-pyrosequencing to study the roots and leaves of the salt-tolerant cultivar under salt stress [79], only 24 genes in the roots (15 up- and 8 downregulated) and 70 in the leaves (56 up- and 14 downregulated) were differentially expressed. Among them, the typical responsive transcription factors were reported (*AP2/ERF*, *NF-Y*, *JERF*, *HMG*, and *GRAS*), ABA-related transcripts, a vacuole-type H^+^-ATPase, a Na^+^/H^+^ antiporter to accumulate Na^+^ in the vacuole, and the plasma membrane antiporter SOS1 (Table 6). The presence of solute transporters and enzymes in the synthesis of osmoprotectants corroborates that salt stress in olive trees triggers an osmotic response at early stages.

Very recently, a multi-omics approach studied the salt priming mechanism in the roots and leaves of the salt-sensitive cultivar ‘Chondrolia Chalkidikis’ [89]. The study revealed major differences between primed and non-primed tissues mainly associated with hormone signalling, carbohydrate metabolism, and defence-related interactions, confirming that salt priming involves reprogrammed transcriptional responses, with the accumulation of ion binding proteins (such as the ATP-dependent zinc metalloprotease (FTSH2) and calcium-transporting ATPase 9 (ACA9), Table 6) in primed roots as a sign of salinity adaptation by ion exclusion mechanisms. They also found that priming the roots increased protein levels of KTI2, a Kunitz trypsin inhibitor (Table 6), contributing to the regulation of cell death already mentioned in Section 3.5.2. But the most interesting fact was that PTMs seem more relevant than transcriptional reprogramming for the salt acclimation of olive trees.

As a general conclusion, olive trees under salt stress are reprogrammed to express an early osmotic response, but the number of genes that significantly change their long-term expression in sensitive and tolerant cultivars is surprisingly low, which is in agreement with findings in citrus trees [170]. More comparative genomics studies with more cultivars should be undertaken to understand the complete range of olive tree responses to salt stress.

### 4.4. Combining Salt and Drought Stresses

In the Mediterranean basin, drought stress is becoming a salt-concurrent threat, resulting in important agricultural losses. Although olive tree is able to cope with water and salt stresses, drought and salt stress alter the olive tree capacity and potential yield in an irreversible way if their severity is very intense [39]. When fresh water is limited or absent due to drought stress, saline irrigation of salt-tolerant olive trees can produce profitable agronomic results, although they were worse than irrigation with good-quality water in the absence of drought stress [54]. Both stresses start by a characteristic osmotic rebalance, and when salt-sensitive cultivars such as Chétoui are grown under this adverse combination of stresses, they develop changes in plant growth (very apparent reduction in aboveground organs), oxidative damage (compensated with an increase in carotenoids), increased osmolyte accumulation in leaves (mainly soluble sugars), lignin accumulation, and reduced photosynthetic performance and nitrogen assimilation [103]. A transcriptomic study using five plant species including olive tree [171] showed that, under concomitant drought and salt stresses, 39 differentially expressed genes based on *Arabidopsis* orthologues were commonly regulated (23 upregulated and 16 downregulated). These genes are involved in defence response, drug transmembrane transport, and metal ion binding. The authors proposed that these genes can be potential targets for developing cultivars with enhanced tolerance to both drought and salt.

### 4.5. Genetic Engineering for Salt Tolerance

Olive cross-breeding began in the second half of the twentieth century and currently represents the most promising strategy to provide farmers with new, well-adapted cultivars or rootstocks [172]. The new released olive cultivar Zeitoun Ennour, issued in 2021 in Tunisia after 28 years of genetic breeding to obtain better fatty acid composition, is a successful example of salt-resistant cultivar [64]: its vegetative growth, biomass allocation, and K^+^/Na^+^ ratio are unaffected by salt, and it excludes Na^+^ and Cl^–^ from the aerial part [64]. Since high-throughput gene expression studies in horticultural plants is known to support the physiological findings [173], genomic information from olive trees regarding salt stress would foster new salt-tolerant cultivars or rootstocks by providing candidate genes, genomic regions, and parents that have the potential to be used in breeding. As a result, short-term salt adaptation was obtained with the genetic engineering of model and crop plants involving ion transporters mentioned in Section 3.4 (comprehensively reviewed in [174]). Studies using transgenic plants overexpressing transcription factors, such as DREB6 [106] or SNAC1 [175], among others, or even miRNAs [139], also end up as salt-tolerant plants (comprehensively reviewed in [176]). But the effectiveness assessment of a transgene on crop salt tolerance requires testing in the field, and only a few promising studies in the field involving transporters have been described (*Triticum aestivum* transformed with *AtHNX1*, *Zea mays* transformed with *OsHNX1*, and *Hordeum vulgare* transformed *AtAVP1*, a vacuolar H^+^-translocating pyrophosphatase) [174]. On the contrary, transgenics involving regulatory elements or effectors remain elusive in the field [176].

Regarding other non-transporter proteins, transgenic *Arabidopsis* and rice overexpressing SOD or other antioxidant enzymes switch to salt tolerance and produce seeds in the presence of 100 mM NaCl [24]. In the case of olive tree, an increased tolerance to salt and drought stresses was observed when the *SCR1* gene encoding a Ks-type dehydrin (Table 6), which was known to be involved in salt tolerance, was overexpressed in tobacco transgenic plants [169]. Transgenic olive cultivars were also envisaged [177]: the Italian transgenic olive cultivars Canino (salt-tolerant) and Sirole (salt-sensitive) overexpressing a tobacco osmotin gene were compared to control non-transgenic plants after four exposures to 200 mM NaCl [178]. Stunted growth and ultimate leaf drop were observed in both cultivars but not in the transgenic lines, suggesting that the S assimilation pathway plays a key role in the adaptive response of olive trees under salt stress conditions.

Unfortunately, genetically modified organisms (GMOs) are difficult to bring to orchards, particularly in Europe, due to the restrictive regulations against GMOs [179]. More hope could be placed on CRISPR-modified plants [180] after recent resolutions [181,182].

## 5. Metagenomics of Salinised Olive Grove Soils

### 5.1. Distribution of Plant-Associated Microorganisms

Plant-associated microorganisms consist of many bacteria, fungi, and archaea taxa, potentially interacting with one another. They live either inside the plant tissue (endophytes) or on the surface of plant organs (epiphytes) in both the aerial part (phyllosphere or ‘aboveground’ microbiota) and the root region (the ‘belowground’ microbiota). The belowground microbiota, considered the key hotspot for plant–microbe interactions, is usually divided into four compartments: the bulk soil, the rhizosphere (outside the root), the rhizoplane (the root surface), and the endosphere (inside the root), with each compartment sustaining distinctive microbial communities. The compartments are not isolated since some endophytes are vertically transmitted to reach the endosphere of the phyllosphere [183]. This was also described in olive tree, where the phyllosphere seems to come from the soil and reaches the aerial plant parts through xylem sap [184], while protecting the plant from diseases and stresses [185]. Plant root-associated microbiota host a reservoir of beneficial microorganisms that improve the growth, nutritional status, development, and fitness of the host plants, including tolerance to drought and salt stresses. These enhancements are achieved directly by improving nutrient assimilation and protection against the stressing factor [186,187,188] and indirectly by shaping agronomically relevant traits [189,190].

### 5.2. Olive Grove Microbiota

The soil microbial composition of olive groves varies depending on the genotype and the studied plant compartments [191,192,193,194] probably by means of root exudates [195] and seems to be modulated by other factors like edaphic conditions and soil management [184,196,197], as well as domestication [198]. Microbiota in olive tree roots across different seasonal patterns are highly stable in taxa composition except in cold-susceptible genotypes [189], although a vast amount of novel diversity remains to be discovered in olive grove soils [193].

*Proteobacteria, Actinobacteria, Firmicutes*, and *Bacteriodetes* dominate the olive tree bacterial microbiota, and in many cases, *Planctomycetes, Acidobacteria, Myxococcota, Verrucomicrobiota*, *Gemmatimonadetes*, and Betaproteobacteria are also relevant [189,191,198]. It seems that the presence of Proteobacteria in the soil is an indicator of a high soil nutrient content [184]. Proteobacteria and its class *Grammaproteobacteria*, *Firmicutes*, and *Actinobacteria* are also highly represented in the olive tree phyllosphere [191,198] while Bacteroidetes are more variable [191,199]. Going deeper into the taxonomy level, the most common genera in soil, leaves, and xylem sap [184] are *Alphaproteobacteria* (more abundant in soil and leaves), Actinobacteria, *Sphingobacteria*, *Bacilli* (more abundant in xylem), *Gammaproteobacteria*, and *Betaproteobacteria*. The most abundant classes in the root endosphere are *Alphaproteobacteria*, *Gammaproteobacteria*, and *Deltaproteobacteria*, belonging to Proteobacteria, with (i) *Actinophytocola* spp., being especially relevant for olive tree fitness and health; (ii) Streptomyces and *Pseudonocardia*, all belonging to *Actinobacteria*, the most abundant genus in the rhizosphere; and (iii) α-Proteobacteria, Rhizobium, *Sphingomonas*, *Sphingomonas* (all Proteobacteria), and *Acinetobacter*, which are the most abundant in soil [192,193,200].

The potential of fungal symbionts in improving plant growth and productivity in saline soils was widely recognised and confirmed [201,202]. Benefits come from the exchange of nutrients, signalling molecules, and protective chemical compounds [203], including olive tree [204], motivating authors to propose that by administering mycorrhiza to plants, they can absorb more soil nutrients and water, reducing fertiliser and other agricultural inputs. Moreover, the presence of fungal symbionts also protects against diseases and reduces weeds and soil erosion [203,205], making them essential for sustainable oliviculture practices in increasingly saline landscapes. Minor mycobiota differences were found between the olive tree root endosphere and the soil rhizosphere. There are also significant differences between the above- and belowground communities [192,194,206], with a predominance of *Basidiomycota* and inconsistent data regarding *Ascomycota*, with four genera (*Canalisporium*, *Macrophomina*, *Aspergillus*, and *Malassezia*) constituting the belowground fungal core [189,191,198]. *Sordariomycetes* are also the most abundant class in olive tree fruits and roots [192,206].

The presence of Archaea is not described in many metataxonomic studies, but there are reports describing *Thaumarchaeota*, *Crenarchaeota*, and *Euryarchaeota* in the root and leaf endosphere, indicating that olive cultivars select specific groups of Archaea as endophytes [196,198].

### 5.3. Decrease in Microbiota Biodiversity Due to Agricultural Practices

Recent studies support the idea that plant microbiota are a consequence of millions of years of co-evolution, where plants likely seek cooperation with microorganisms by means of chemical stimuli as a kind of ‘cry for help’ strategy to fight stresses [186]. It was reported that *Proteobacteria*, *Actinobacteria*, and *Bacteroidetes* are the most abundant bacteria, and that *Ascomycota* and *Basidiomycota* are the most abundant fungi in three olive groves with differences in soil and climate (Jaén, Córdoba, and Málaga). Moreover, *Actinophytocola*, *Streptomyces*, and *Pseudonocardia* are the most abundant bacterial genera in the olive tree root endosphere in the Spanish germplasm bank, with *Actinophytocola* being the most prevalent genus by far [192].

In other levels of co-evolution, the beneficial effects of the natural above- and belowground microbiota are threatened by human agricultural practices related to pesticides, fertilisers, or tillage since they tend to decrease microbiota biodiversity [207]. This prompted the proposal of a defence biome [186] to guide the design and construction of beneficial microbial synthetic communities. Low-quality soils of olive groves result in the endosphere and rhizosphere being highly affected by the agronomic practices in a cultivar-dependent way [184,191,208,209]. As a result, olive growers have been forced to use agrochemicals that arouse negative effects on the environment and ecosystem biodiversity to promote olive tree growth, control plant pathogens, and increase the nutritional value and quality of olive products. Hence, new alternative eco-friendly practices aiming to increase olive tree yield and health by controlling biotic threats are required, involving the use of beneficial microorganisms that can also decrease agricultural inputs and improve soil structure and fertility [210].

### 5.4. Olive Tree Microbiota Changes Under Salt Stress

It is known that salt stress affects microbial diversity, community structure, and functions in many crops [211]; bacterial diversity decreases under high salt, whereas the response of fungi to this stress is more complex. In general, soil salinisation leads to the replacement of salt-sensitive taxa by salt-tolerant ones [212], resulting in a salt-tolerant microbial community that may sustain and even promote plant growth [34,213]. Microbiota contributes to plant growth in salinised soils by means of biofilm formation, extracellular polymeric substance production, nitrogen fixation, phytohormone production, nutrient uptake promotion, and homeostasis [34,212]. Other microorganisms induce the expression of salt-responsive genes via the action of transcription factors, as well as by post-transcriptional modifications and PTMs [212].

Moderate levels of salt stress can favour bacterial diversity of soil microbial communities instead of their rapid replacement by salt-tolerant fungi at high salt levels, suggesting that the interaction of the epigenetic adaptation process in the plant and the microbiota occurs under high salt stress but not under mild salt stress. However, in olive cultivar Arbequina, a moderate salinity (100 mM NaCl) treatment reshaped the microbiota composition [214]. In fact, there is a shift in the phyllosphere bacterial communities, with a significant increase in the *Burkholderia* and *Ralstonia* taxa (both belonging to *Burkholderiales*) and a decrease in their antagonistic counterpart *Pseudomonas* (a genus that improves salt tolerance in plants [211,215,216] and is the main taxon of native olive leaf microbiota [199]). In summary, there is much evidence that salt-tolerant bacteria can protect plants (including olive tree) from salinisation and would improve productivity and yield under salt stress regimes.

### 5.5. Soil Amendments and Salt Stress

Restoring the quality of degraded soils by salinisation is a challenging task, especially in arid and semi-arid regions, even though soil health restoration has recently gained increased attention to recover global biodiversity and habitat (loss driven by human activities and climate change) and rebuild ecosystem resilience and sustainability [217]. Biostimulants (materials of biological origin that enhance plant growth or development when applied to the soil), as well as chemical compounds (such as biochar [218], melatonin [219], silicon, or ascorbic acid [143]), have been used in crop groves (including olive tree) as cost-effective soil amendments to increase plant yield, stress tolerance, and even soil health and fertility [220]. In some cases, the amendments induce microbial cells to counteract the osmotic stress with the production of osmolytes that mitigate the effects of salinisation [221], demonstrating close crosstalk between plants and the inhabitant microbiota favoured by the amendment [200]. Chemical amendments have also been used with the olive tree cultivars Koroneiki and Mastoidis, showing that salt toxicity symptoms are partially reverted in the presence of 100 mM K^+^ [222]. Other chemical compounds with CaCl_2_, MgCl_2_, H_2_SO_4_ (the three added to the saline irrigation water), and gypsiferous material (added to the soil) are able to maintain tree productivity, although a high Mg^2+^ concentration leads to some nutrient imbalances [223]. Adverse effects of salinisation are partially ameliorated by proline in ‘Arbequina’, ‘Arbosana’, and ‘Koroneiki’ [65]. The use of compost, olive mill wastewater, and legume cover crops is also beneficial for ‘Chemlali’ irrigated with saline water under semi-arid conditions [224]. Since Zn^2+^ is required for ROS detoxification, its supplement improves the antioxidative defence and increases the salt tolerance in many plants, including the olive cultivars Frantoio and Conservolea, by alleviating oxidative injuries [225]. Hence, chemical amendments can be beneficial to olive fruit production when other salt toxicity mitigation strategies are not available.

When the described conventional strategies cannot alleviate the ion toxicity, are difficult to implement, or are harmful to soil health [226], enhancing biodiversity to promote pedosphere processes appears as an alternative. This is why the manipulation of plant microbiota is an emerging practice to mitigate salt stress, even in olive tree [227]. Several studies have demonstrated that, when inoculated to soils, growth-promoting rhizobacteria (PGPR) from the genera *Klebsiella*, *Streptomyces*, *Pseudomonas*, *Agrobacterium*, *Bacillus*, *Enterobacter*, *Stenotrophomonas*, *Rhizobium*, and *Ochromobacter* avoid the over-accumulation of Na^+^ and maintain ion homeostasis under salt stress, resulting in an improved crop yield [228]. Several successful studies have identified soil and rhizosphere bacteria (and, to a lesser extent, fungal taxa) as potential plant growth-promoting microorganisms (PGPM) able to reduce the impact of salt stress in crops [211,216,229]. Regarding olive tree, *Glomus mosseae* was described as the most efficient fungus for reducing the detrimental effects of salt in olive plantlets, concomitant with an enhanced K^+^ concentration in the plant [204], with its inoculation in olive tree pot soils being a current practice in nurseries. More recently, it was demonstrated that *Bacillus* G7 stimulates the adaptive response to high-salinity conditions by increasing the photosynthetic efficiency and the content of osmolytes and antioxidants, improving carbon fixation and water use efficiency, and upregulating ion homeostasis-related genes and ABA synthesis [230]. The multitarget and reprogramming mechanisms observed revealed the multiple mechanisms affected by *Bacillus* G7 to improve the adaptation of olive tree to salt stress.

## 6. Bioinformatics of Salt Stress in Olive Tree

### 6.1. From Olive Tree Transcriptome to OliveAtlas

The first sequencing studies concerning olive trees were performed in the 21st century [231]. The high-throughput sequencing data were later collected to produce the first olive tree transcriptome [232] and then a database for the transcriptome of olive tree reproductive tissues [233]. The authors of the present review have also developed olive-focused bioinformatic tools that are useful for studying salt stress in citrus [170]: (1) RSeqFlow [234] to study differential gene expression that is currently in use to study salt and drought stresses; (2) a comprehensive list of *Arabidopsis* orthologues to ‘Picual’ [234] adapted for functional analyses based on the clusterProfiler v4.14 package (https://github.com/bullones/FunRichR; accessed on 10 March 2025); (3) an R markdown pipeline for new olive allergen detection (https://github.com/bullones/AllergenDetectR; accessed on 10 March 2025); and (4) an R markdown pipeline to detect gene co-expression based on the WGCNA v1.73 package (https://github.com/Javiersdr/Co-expression_analysis; accessed on 10 March 2025).

The continuous growth of high-throughput sequencing data, the availability of olive genome sequences (Table 5), and the continuous advances in artificial intelligence algorithms require new bioinformatics approaches. It is time for expression atlases that (1) integrate expression data from different sources such as RNA-seq, proteomics, metabolomics, and expression microarrays; (2) precompute many analyses using standardised methods; and (3) incorporate multiple interactive visualisation methods to explore and analyse expression data. Successful cases of atlases in plants are the Tomato Expression Atlas (TEA) [235], the atlas of the Norway spruce needle seasonal transcriptome [236], and many plants in ePlants (http://bar.utoronto.ca; accessed on 10 March 2025) at BAR [237], PEATmoss for *Physcomitrium patens* [238], and MangoBase for *Mangifera indica* [239]. The OliveAtlas (https://www.oliveatlas.uma.es/; accessed on 10 March 2025) for the *Olea europaea* subsp. *europaea* cultivar Picual was recently published [240], integrating information on 70 RNA-seq experiments organised by datasets that include, among others, the response to biotic and abiotic stresses. Our ongoing multi-omics data investigating responses to salt stress of four olive cultivars (Figure 2) will soon be integrated into the OliveAtlas. Expression data from other research groups, as well as other olive tree genome references, will be integrated and subjected to machine learning approaches to provide clues (i.e., genes and biomarkers) for the selection of the most suitable cultivars tolerating salt stress. Researchers can find the gene expression levels under different environmental conditions without performing expensive, time-consuming experiments. This would help select biomarkers and biosensors (genes, polymorphism/alleles, phenotypes, and even microbiota) that can guide research and agricultural practices to include the desired tolerance traits in new olive rootstocks or cultivars by introgression. Hence, OliveAtlas can be considered the first step towards future smart (or digital) breeding under the current climate change scenario.

### 6.2. Computational Integrative Approaches to Study Salt Stress

Artificial intelligence (AI) techniques are rapidly progressing in many fields of agriculture, encompassing tasks such as counting plants, fruits, or leaves, crop classification, phenotyping, plant disease diagnosis, weed control, and biotic stress prediction [27,241,242,243]. The integration of physiological, biochemical, and molecular markers with AI techniques for predicting plant stress remains an underexplored area [244,245,246]. Promising studies combining image data (source of morphology and phenomics) with genomics datasets in the widest sense [245] have focused on plant phenotyping [241,242], biotic and abiotic damage [243], differentiation of canopy and non-canopy regions [247], and the identification of stressed leaves [241,242]. In particular, machine learning (ML) models have enabled the identification of genes and miRNAs in six types of abiotic stress (salinisation, oxidation, light, heat, cold, and drought) [248,249], also revealing that support vector machines (SVMs) outperform other ML approaches, including deep learning (DL). Yield prediction under salt stress in breeding programmes for alfalfa and many other polyploid crops is another successful use of ML to integrate polymorphisms and genomic context [250,251]. The integration of time-series transcriptome data from various stress types (heat, cold, salt, and drought) using ML has resulted in more accurate stress classification, with high specificity in the discovery of known stress-related genes [252]. Recently, salt stress was tackled with a multi-omics approach in *Fabaceae* plants (which enabled the development of ‘smart farming’ with genome-wide marker-assisted breeding and the removal of deleterious genes [253]) and rice (with the application of AI to obtain accurate predictions [254]). As a result, more and more pipelines and web tools designed for multi-omics datasets are emerging, such as ‘mixOmics’ [255], ‘OmicsIntegrator’ [256], or ‘MiBiOmics’ [257], as well as AI-driven tools that reduce the need for human intervention, such as ‘AutoBA’ [258]. AI tools and databases to feed algorithms have recently been compiled [245]. Hence, multi-omics AI strategies have the potential to revolutionise not only plant stress research but also crop breeding programmes [246,259,260,261].

### 6.3. Machine Learning Studies in Olive Tree

AI, in general, and ML, in particular, are virtually unexplored fields in olive tree. A pioneer study developed an algorithm to detect symptoms of olive quick decline syndrome on leaves infected by *Xylella fastidiosa* from leaf images [262]. The method is able to discover veins and colours that lead to symptomatic leaves. The study also demonstrated that transfer learning (the use of other model plants where more data can be found to obtain models that will then be tested in another organism) of the same disease in other plants can be leveraged when it is not possible to collect thousands of new leaf images in olive trees. A more recent approach is able to detect symptoms of this infection using image processing techniques applied to high-resolution visible and multispectral images [263]. Detection is not only fast but also highly sensitive (98%) and precise (93%). DL has been used to define the development phases of the olive fruit fly to foresee the best times to apply the countermeasures to prevent the outbreak of such a fatal pest [264]. Another study utilised DL for the accurate (89.57%) identification of olive cultivars using a combination of morphological characteristics and ISSR (inter-simple sequence repeats) markers [265]. However, studies about salt stress in olive trees remain to be conducted, in spite of its high potential for elucidating the salt tolerance mechanisms that would foster the selection of salt-tolerant rootstock and cultivars [261], as well as optimal microbiota, for salinised soils.

## 7. Conclusions: Stepping Towards ‘Smart Oliviculture’

There is an urgent need to combat soil salinisation. Soil amendments (including microbiota) and salt-tolerant cultivars and rootstocks seem to be realistic approaches for salt stress mitigation to restore the microbial community and the agricultural use of salinised lands. Non-adaptive responses to salt trigger the mechanisms of PCD (Figure 1), making the metacaspases involved in PCD and autophagy promising targets for new, CRISPR-based biotechnological strategies that can enhance salt tolerance (and likely other stresses) in crop cultivars, at least in laboratories. The economic and nutritional interest in olive oil, the abundance of olive groves in lands affected by soil salinisation, and the resilience of olive tree to ion exposure make it an interesting model of salt stress resilience. Efforts should be dedicated to determining whether salt-tolerant olive cultivars (Table 1) may require less fertiliser due to their better nitrogen uptake (which, in addition, would diminish nitrate leaching into groundwaters). The use of salt-tolerant olive rootstocks or cultivars would save time since there would be no need to develop new salt-tolerant cultivars. Priming rootstocks, olive seedlings, or even soils with PTM inducers or organic amendments are promising alternatives for existing olive groves. Future breeding programmes aiming to produce new salt-tolerant cultivars or rootstocks would be assisted by the emerging genome information on olive trees (Table 5) and the differentially expressed genes identified after long-term salt treatments (Table 6).

Although it seems that at least cell wall thickening, ion exclusion, and antioxidant response including a rise in osmoprotectants are concomitantly required for salt tolerance in olive tree, more multi-omics studies contemplating PTMs and microbiota should be accomplished in olive tree to achieve a complete picture of salt stress. This analysis should take advantage of AI, particularly ML and SVMs, to combine physiology, biochemistry, phenomics, and genomics datasets and even remote agrisensors in a kind of ‘smart oliviculture.’ Data integration in an open science resource (for example, by means of the OliveAtlas) would provide scientists and growers with a powerful method of finding information about basic and applied knowledge related to soil salinisation and the consequent olive damage. This is expected to stimulate sustainable and resilient ‘smart oliviculture’ by (1) providing clues to manage olive cultivars and groves to cope with salinisation to maintain or increase productivity and (2) providing information about the possibility of maintaining current cultivars irrigated with poor-quality water while waiting for new salt-tolerant rootstocks. This is not speculation since companies are offering in-field remote sensors that provide real-time data on factors affecting crop health and overall yield.

## Figures and Tables

**Figure 1 biology-14-00287-f001:**
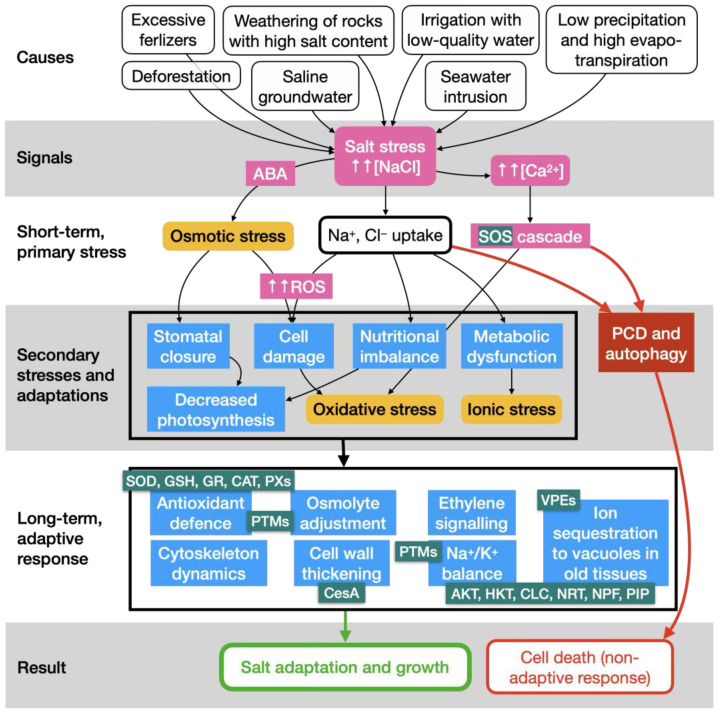
Integrative scheme showing main causes of natural and anthropogenic soil salinisation (white boxes), their relationship to salt adaptation and growth (green arrow and box) or non-adaptive response (red arrows and boxes), and associated stresses (orange boxes). Arrows depict known cause–effect relations. Nain signals are represented in pink boxes, adaptive processes in blue boxes, and biochemical activities in overlapping, teal-coloured text background. Alternating white and grey strips help to differentiate different stages of salt stress. ABA: abscisic acid; CesA: cellulose synthase; ROS: reactive oxygen species; SOS: salt overly sensitive; PCD: programmed cell death; PTMs: post-translational modifications; VPEs: vacuolar processing enzymes; SOD, GSH, GR, CAT, and PXs: antioxidant enzymes (see Section 3.1); AKT, HKT, CLC, NRT, NPF, and PIP: transporters (see Section 3.2).

**Figure 2 biology-14-00287-f002:**
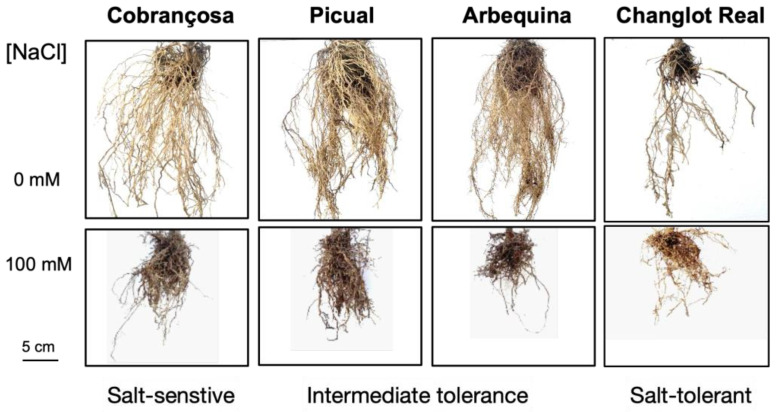
Effect of salt stress at 100 mM NaCl for four months on roots of olive cultivars with different sensitivities to salt. Left cultivar is salt-sensitive ‘Cobrançosa’, two middle cultivars have intermediate tolerance (‘Picual’ and ‘Arbequina’), and cultivar on right is salt-tolerant ‘Changlot Real’. Clear decrease in root mass and length, as well as evident change in colour, can be observed in salt-sensitive and intermediate tolerant cultivars, with respect to control experiment where no NaCl was added; changes in salt-tolerant cultivar are less evident.

**Table 5 biology-14-00287-t005:** *Olea europaea* genomes sequenced to date.

Subsp.	Cultivar	Last Version	Scaffold N50 (Mb)	Genome Size (Gb)	%GC	Gene Models	#AC	Reference
*europaea*	Farga	OLEA9	0.73	1.38	33.8	56,349	GCA_902713445.1	[155,156]
*sylvestris*	Wild type	O_europea_v1	12.57	1.48	31.9	47,911	GCA_002742605.1	[44]
*europaea*	Picual	Oleur0.6.1	0.41	1.68 ^a^	33.8	79,667	PRJNA556567 ^b^	[157]
*europaea*	Arbequina	Olive	42.6	1.30	34.3	53,518	GWHAOPM00000000 ^c^	[158]
*europaea*	Leccino	T2T_Lec	54.85	1.28	35.9	70,138	GWHEUUU00000000.1 ^c^	[159]
*europaea*	Leccino	-	45.86	1.43 ^a^	36.87	67,103	PRJNA1197712	[160]
*europaea*	Frantoio	-	1.78	1.18 ^a^	35.81	59,777	PRJNA1197703	[160]
*cuspidata*	-	JYM_FINAL	52.68	1.38	34.7	46,904	CNP0002655 ^d^	[161]
*cuspidata*	KM Yaf	YAF_Ocus_V1	50.2	1.20	35.3	43,511	GCA_023089605.1	[162]

^a^ Assembled as scaffold draft, not chromosomes. ^b^ Assembly downloadable from https://genomaolivar.dipujaen.es/db/downloads.php (accessed on 30 January 2025). ^c^ Assembly downloadable from the Genome Warehouse (GWH) https://ngdc.cncb.ac.cn/gwh/ (accessed on 30 January 2025). ^d^ Assembly downloadable from the China National GeneBank DataBase (CNGBdb) https://db.cngb.org/ (accessed on 30 January 2025).

**Table 6 biology-14-00287-t006:** Summary of genes involved in salt stress response in olive tree.

Gene Name	Description	Function	Response	Cultivar	Method	References
PPO	Polyphenol oxidase	Tyrosine metabolism	Up ^3^	Chondrolia Chalkidikis (salt-sensitive)	qPCR, multi-omics	[88,89]
hisC	Histidine decarboxylase					
F3H	Anthocyanidin 3-O-glucosyltransferase	Flavonoid metabolism				
FNSII	Flavone synthase II					
CA4H	Cinnamate 4-hydroxylase					
PLRTp2	Pinoresinol-lariciresinol reductase 2	Lignan metabolism				
GTF	Glucosyl transferase flavouring	Secoiridoid metabolism				
DRP	Desiccation-related protein PCC13	Cellular response to salt				
PP1	Peptidyl prolyl cis/trans isomerase	Protein folding and brassinostroid response				
G80	8-Hydroxygeraniol dehydrogenase	Oleuropein biosynthesis				
EREBP	Ethylene-responsive transcription factor 1B	Plant development and stress signalling				
KTI2	Kunitz trypsin inhibitor 2	Control of cell death				
FTSH2	ATP-dependent zinc metalloprotease	Thylakoid formation and removal of damaged D1 in photosystem II				
ACA9	calcium-transporting ATPase 9	Transport of cytosolic Ca out of the cell or into the organelle				
FAD6	Fatty acid desaturase	Conversion of oleic acid (C18:1) to linoleic acid (C18:2)	Down ^3^	Leccino (salt-sensitive)	qPCR	[51]
SAD1	Stearoyl-acyl carrier protein desaturase	Conversion of stearic acid (C18:0) to oleic acid (C18:1)	Up ^3^			
NHX7	Sodium/hydrogen exchanger (NHX)	Ion transport across cellular membranes	Up ^1^	Royal de Cazorla (salt-tolerant) Picual (moderately tolerant) Fadak86 (salt-sensitive)	qPCR	[85]
P5CS	Δ¹-pyrroline-5-carboxylate	Proline biosynthesis				
RD19A	Responsive to dehydration/dehydrin	Stabilising proteins and membranes under water-limited conditions				
PetD	Electron transfer between cytochrome b6f complex and photosystem I	Component involved in photosystem complexes within chloroplasts specifically associated with electron transport chains during photosynthesis				
V-H^+^-ATPase subE	Vacuole-type H^+^-ATPase subunit E	Pumping protons (H^+^ ions) across cellular membranes	Up ^2^	Frantoio (salt-tolerant) Leccino (salt-sensitive)	qPCR	[137]
NHX	Sodium-hydrogen exchanger	Vacuolar Na^+^/H^+^ antiporter	Up ^2^			
P-ATPase11	P-type ATPase,	Pumping protons (H^+^ ions) across cellular membranes	Up ^1^			
P-ATPase8	P-type ATPase,					
SOS1	Salt overly sensitive 1	Plasma membrane Na^+^/H^+^ antiporter				
NHX7	Sodium/hydrogen exchanger (NHX)	Ion transport across cellular membranes	Up ^1^	Royal de Cazorla (salt-tolerant) Picual (moderately tolerant) Fadak86 (salt-sensitive)	qPCR	[85]
P5CS	Δ¹-pyrroline-5-carboxylate	Proline biosynthesis				
RD19A	Responsive to dehydration/dehydrin	Stabilising proteins and membranes under water-limited conditions				
PetD	Electron transfer between cytochrome b6f complex and photosystem I	Component involved in photosystem complexes within chloroplasts specifically associated with electron transport chains during photosynthesis				
PIP1.1	Aquaporin	Water transport across cell membranes	Up ^1^	Royal de Cazorla (salt-tolerant) Picual (moderately tolerant) Fadak86 (salt-sensitive)	qPCR, MSAP ^4^	[84]
PetD	Electron transfer between cytochrome b6f complex and photosystem I	Component involved in photosystem complexes within chloroplasts specifically associated with electron transport chains during photosynthesis				
PI4Kg4	Phosphatidylinositol 4–kinase gamma	Phosphatidylinositol metabolism/lipid signalling pathway				
XylA	Xylose isomerase	Conversion of D-xylulose into D-xylose				
MO1	Monooxygenase 1	Addition of an oxygen atom into organic substrates/synthesis of secondary metabolites	Down ^1^	Picual (moderately tolerant) Nabali (moderately tolerant)	qPCR	[168]
STO	Salt tolerance protein	Transporter or chaperone to facilitate ion exchange				
PMP3	Proteolipid membrane potential modulator	Modulation of membrane potential across cellular membranes				
USP2	Universal stress protein	Molecular chaperone				
AP-4	Adaptor protein complex 4 medium mu4 subunit	Vesicle trafficking within cells				
WRKY1	WRKY1 transcription factor	Transcription factor				
CCX1	Cation calcium exchanger 1	Cation/proton antiporter	Up ^1^			
KT2	Potassium transporter 2	Potassium channel				
AP2/ERF	Apetala2/Ethylene response factor	Transcription factors	Up ^1^	Kalamon (salt-tolerant) Chondrolia Chalkidikis (salt-sensitive)	RNAseq	[78,79]
NF-Y	Nuclear factor Y					
JERF	Juvenile ethylene response factor					
GRAS	Gibberellic acid receptor-like family					
bZIP	Basic leucine zipper					
HMG	High mobility group	Gene transcription regulation DNA replication, repair mechanisms, and chromatin remodelling				
SGT	Putative small glutamine-rich tetratricopeptide repeat-containing protein	Protein–protein interactions				
RPS6	40S Ribosomal protein	Protein synthesis and signal transduction pathways				
ART/PARP	NAD+ ADP-Ribosyltransferase	Signal transduction pathways				
ANXA4	Annexin A4	Membrane stabilisation and vesicle trafficking				
XET	Xyloglucan endotransglycosylase	Elongation by modifying cell walls				
UGE	UDP-Galactose epimerase	Carbohydrate metabolism				
SCR1	KS-type dehydrin	ROS-reducing activity that provided salt and drought tolerance	Up	Transgenic tobacco		[169]

^1^ Up/down in tolerant vs. sensitive cultivar during salt stress. ^2^ Up/down in both tolerant and sensitive cultivar during salt stress. ^3^ Up/down during salt stress. ^4^ MSAP: methylation sensitive amplification polymorphism.

## Data Availability

This review does not provide new data.

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
