# Peer review of "Multi-Omic Advances in Olive Tree (Olea europaea subsp. europaea L.) Under Salinity: Stepping Towards ‘Smart Oliviculture’"

_biology, 2025, doi:10.3390/biology14030287_

Round 1
Reviewer 1 Report (New Reviewer)
Comments and Suggestions for Authors
Salt stress affects the plant productivity and is a global issue in Agriculture. Soil affected by salinity increases every year due to the poor quality of the water used for irrigation. Hence working on Salt stress is significantly important area. The authors reviewed the advances in genomics for enhancing the salt tolerance in Olive trees. The authors clearly expressed the need of this review and provided a good introduction. Although it is well framed, some revision is required.
- Check spellings. In figure 1 under strip ‘causes’ – “Excessive ferlisers”? Is it fertilizers?
- In references, check the details very carefully. For example, the year of the publication of reference 46 is 1997. Not 2010; https://doi.org/10.1002/9780470650660.ch6.
- Is the figure 2 is from the current authors? Although it was kept after “stalk length, leaf surface, dry weight, and root length”, the figure only shows roots. Based on the root length they all looks more similar. The roots from salt tolerant variant shows brownish roots in the control condition and comparatively low growth. Why? After how many days these results were observed? How the treatment was given? If already published provide the reference.
- In line 350 to 357, gene names not expanded. Similarly check throughout the manuscript.
- It will be better if refences in the table were provided in the last column (for all tables)
Comments on the Quality of English Language
The language needs to be revised for better readability and clarity.
Author Response
We would like to acknowledge the positive comments of reviewer #1 as well as the detection of flaws requiring correction.
Comment 1
Check spellings. In figure 1 under strip ‘causes’ – “Excessive ferlisers”? Is it fertilizers?
Response
Thank you for your comment. The manuscript was written in Britsh English, where are admitted the forms with ‘s’ and ‘z’ of many words. However, ‘fertilizer’ is correct here but not ‘fertiliser’. We have corrected this in Fig 1 and the main text, and the modifications are indicated by track changes.
Comment 2
In references, check the details very carefully. For example, the year of the publication of reference 46 is 1997. Not 2010; https://doi.org/10.1002/9780470650660.ch6.
Response
Thank you very much for revealing this error. We changed the year.
Comment 3
Is the figure 2 is from the current authors? Although it was kept after “stalk length, leaf surface, dry weight, and root length”, the figure only shows roots. Based on the root length they all looks more similar. The roots from salt tolerant variant shows brownish roots in the control condition and comparatively low growth. Why? After how many days these results were observed? How the treatment was given? If already published provide the reference.
Response
Regarding this comment, the figure was composed from our experiments and has not been published before, and this is why no reference was given and cannot be given. The figure is only illustrative purposes after 4 month exposure to 100 mM NaCl (this was stated in the first line of the figure legend) to show the morphological changes described for roots by other authors in previous articles. So, since the manuscript is a review and not an experimental analysis, the figure is to be understood as merely illustrative. Concerning the root length and colour, you can see that the roots for each cultivar without salt (upper panels) and under salt stress (lower panels) are different: roots are clearly shorter in the presence of salt for all cultivars, and are darker under salt stress, even if the initial root seems more brownish (as the referee remarked in Changlot Real). We do not know what is the reason of the colour (maybe phenolic compounds?), we have not studied that, and this phenotype is not among the goals of our review.
Comment 4
In line 350 to 357, gene names not expanded. Similarly check throughout the manuscript.
Response
Thank you for pointing this out. Genes known to be involved in salt stress in olive are gathered in Table 6. Genes mentioned by the referee are from model species and require expansion. Hence, the meaning of their names has been included for those genes mentioned by the referee as well as those appearing in lines 365-366. Track changes highlight the name expansions introduced.
Comment 5
It will be better if refences in the table were provided in the last column (for all tables)
Response
We appreciate the commentary of the referee, but the reason why Tables 1 to 4 have references in the second column is to provide internal homogeneity: in Table 4, the reference indicates which articles are supporting the level of salt tolerance, and it is in this position since the most important information of these tables is related to the level of tolerance. Since rows in Tables 1-3 are all for the same level of tolerance, references are in the same position to justify why several cultivars present more than 1 row.
Reviewer 2 Report (New Reviewer)
Comments and Suggestions for Authors
The authors have done a commendable work in the the review manuscript, with an excellent flow of information starting from the basics, available varieties, mechanisms and future perspectives with a well prepared illustrations using appropriate tables and figures.
The references are very extensive and exhaustive in the area of research.
I have only very MINOR comments to offer for its further improvement:
- Figure 1: Correct the spelling of 'fertilisers'
- Table 1, 2 & 3: The title could be revised with more details
- Check the correctness of MgSO4
- Line 249: Revise as '28 years of breeding'
Author Response
We would like to acknowledge the very positive comments of reviewer #2 as well as the minor flaws that require corrections.
Comment 1
Figure 1: Correct the spelling of ‘fertilisers’
Response
Thank you for your comment. The remarks is appropriate and we changed all ‘fertiliser’ in Figure and text to ‘fertilizer’. Modifications were highlighted.
Comment 2
Table 1, 2 & 3: The title could be revised with more details
Response
Thank you for highlighting this issue. Title of these tables has been completed with “based on analytical data or common usage”.
Comment 3
Check the correctness of MgSO4
Response
Thank you for pointing this error out. Although MgSO4 is correct, we have mended many NaSO4 to Na2SO4 in Tables 1 to 4, while it was correct in the text.
Comment 4
Line 249: Revise as ‘28 years of breeding’
Response
Following this comment, we have removed this information from line 249 since it is not relevant there, but we maintain it at the beginning of Section 4.5 since the genetic improvement project leading to Zeitoun Ennour was initiated in 1993 or 1994 (depending on the reference consulted) and this cultivar was described for the first time in 2021 (https://doi.org/10.46265/genresj.FIQJ8274).
Reviewer 3 Report (New Reviewer)
Comments and Suggestions for Authors
The review “Advances in genomics to improve salt tolerance in olive tree (Olea europaea subsp. europaea L.): stepping towards “smart oliviculture”” is devoted to a comprehensive examination of salt tolerance in olive trees. The response of olive trees to salt stress, the manifestation of salt tolerance in different cultivars, mechanisms of adaptation to stress, changes in microbiota under the influence of stress, genomic and bioinformatic studies related to this topic are discussed in detail. The manuscript is easy to read, the material is presented logically and consistently and is accompanied by a large number of examples and references to literature. The work may be of interest to a wide range of readers, especially those involved in the study of stress resistance in perennial and woody plants. However, there are a few remarks.
- The current title does not fully reflect the content of the review. The work is devoted not only to genomic research. Perhaps it is worth considering changing the title of the manuscript.
- Figure 1. In the upper left corner there is a typo in the word "fertilizers".
- Lines 161-166. Data is given for Spain, but is there data for other countries? Can the data given for Spain be extrapolated to all regions where olive trees are grown?
- Is there any comparison - which strategy is more profitable - grafting salt-sensitive shoots onto salt-tolerant roots, or using salt-tolerant cultivars?
- Lines 325-326. The sentence needs to be reformulated - in its current form it sounds like only olive trees have ROS levels raised under salt stress.
- Line 411. The explanation of abbreviations is found in the text significantly later than the first use of the abbreviations.
- Perhaps sections 4 and 6 should be placed sequentially one after the other. It is logical that bioinformatics studies follow genomic ones.
- Line 595. The statement "Olive breeding was only initiated in the second half of the twentieth century" sounds implausible, since olive trees are believed to have been domesticated 5500 BC. It is better to clarify (as is written in the paper at reference 178) that cross-breeding began in the second half of the 20th century.
- The logical connection between the sentences on lines 701-705 and the sentence on lines 706-709 is unclear. How does it follow from the information about the main types of bacteria and fungi in olive groves that olive producers were forced to use agrochemicals that affected the ecosystem?
- Lines 718-722 - the sentence seems too cumbersome and difficult to understand. Perhaps it should be split into several sentences or simplified.
- Line 833. The abbreviation ML is explained significantly later than the first mention of the abbreviation.
Author Response
We would like to acknowledge the nice words of reviewer #3 as well as the remarks that require our intervention.
Comment 1
The current title does not fully reflect the content of the review. The work is devoted not only to genomic research. Perhaps it is worth considering changing the title of the manuscript.
Response
Thank you for your comment, that make us change the title to
Multi-omic advances in olive tree (Olea europaea subsp. europaea L.) under salinity: stepping towards “smart oliviculture”.
Accordingly, we changed section “Microbiota changes…” to “Metagenomics studies…” so that sections 4, 5 and 6 are devoted to different omics studies.
Comment 2
Figure 1. In the upper left corner there is a typo in the word “fertilizers”.
Response
We changed ‘fertiliser’ in Figure 1, and also in text, to ‘fertilizer’. Modifications were highlighted.
Comment 3
Lines 161-166. Data is given for Spain, but is there data for other countries? Can the data given for Spain be extrapolated to all regions where olive trees are grown?
Response
We are sorry, but we were not able to find these data for other countries since they are expected to be generated by Government Offices and not scientific papers. We are nearly sure that these data must exist, but published in local languages. Since the goal of our manuscript is not related to agronomic production, we think that the illustration of this aspect in the main producer of olive oil and fruits can be a good case of the world situation.
Comment 4
Is there any comparison - which strategy is more profitable - grafting salt-sensitive shoots onto salt-tolerant roots, or using salt-tolerant cultivars?
Response
Unfortunately, we were not able to find such a comparison in olive tree to provide a response to the reviewer. Works are devoted to look for tolerant rootstocks or to study different levels of salt-tolerance in cultivars, usually from plantlets and not field trees.
Comment 5
Lines 325-326. The sentence needs to be reformulated - in its current form it sounds like only olive trees have ROS levels raised under salt stress.
Response
I am sorry, but the sentence reflects the information that can be found published by Regni et al (2019) and others. Here you are the exact sentences from this publication:
“A major biochemical alteration […] is the production of reactive oxygen species (ROS). […] Under stress conditions, plants can nonetheless develop tolerance […], following a robust production of antioxidant enzymes. […] Indeed a low chlorophyll content in leaves of stressed plants, as observed in the olive plants, is a typical effect of NaCl exposure, associated with an increase of oxidative stress and, at the same time, an increase in ROS scavenging enzymes as a physiological response”.
Comment 6
Line 411. The explanation of abbreviations is found in the text significantly later than the first use of the abbreviations.
Response
Thank you for pointing this out. The meaning of PTM abbreviation was already stated in Section 1.2 and Figure 1 legend, can be found in the Abbreviation appendix of the manuscript, and is profusely explained in next lines, so we decided to maintain it. PCD was already stated in Figure 1 legend and in section 3.4, but in this sentence we agree with the reviewer that it is preferred the expanded name, and we changed it. It is highlighted in the text.
Comment 7
Perhaps sections 4 and 6 should be placed sequentially one after the other. It is logical that bioinformatics studies follow genomic ones.
Response
The referee’s suggestion has sense, but bioinformatics is also be applied to metagenomics. Moreover, multi-omic analysis tools are described in bioinformatics section to integrate information coming from different omics. This is why we placed bioinformatics as a final section. Additionally, bioinformatics acts as bridge from experimental results to a future smart oliviculture, reinforcing the idea that this section must be the last one.
Comment 8
Line 595. The statement “Olive breeding was only initiated in the second half of the twentieth century” sounds implausible, since olive trees are believed to have been domesticated 5500 BC. It is better to clarify (as is written in the paper at reference 178) that cross-breeding began in the second half of the 20th century.
Response
Thank you very much for highlighting this flaw in our manuscript. We have reworded the sentence as follows:
Olive cross-breeding began in the second half of the twentieth century.
Comment 9
The logical connection between the sentences on lines 701-705 and the sentence on lines 706-709 is unclear. How does it follow from the information about the main types of bacteria and fungi in olive groves that olive producers were forced to use agrochemicals that affected the ecosystem?
Response
Thank you very much for highlighting this flaw in our manuscript. The sentences you indicated were not located in the right place since they are breaking the rationale of the consequences of agricultural practices. These sentences were moved to the beginning of the paragraph since they only illustrate what is found in olive groves that can be affected by some agricultural practices. Two different paragraphs are now reflecting these informations.
Comment 10
Lines 718-722 - the sentence seems too cumbersome and difficult to understand. Perhaps it should be split into several sentences or simplified.
Response
The comment of the referee is very appropriate since the sentence repeats the same idea. So it has been shortened to
Microbiota contributes to plant growth in salinated soils by means of biofilm formation, extracellular polymeric substance production, nitrogen fixation, phytohormone production, nutrient uptake promotion, and homeostasis [34,218].
Comment 11
Line 833. The abbreviation ML is explained significantly later than the first mention of the abbreviation.
Response
Thank you for this suggestion. We use now ‘machine learning’ in Section 6.1 so that the first appearance of ML in Section 6.2 includes the explanation.
This manuscript is a resubmission of an earlier submission. The following is a list of the peer review reports and author responses from that submission.
Round 1
Reviewer 1 Report
Comments and Suggestions for Authors
My suggested changes are mentioned below
1. Soil salinization refers to the process of salt buildup while salinity stress denotes the effects of excess salt on crop growth and development. In this context, authors are advised to modify the manuscript title as: ‘Advances in genomics to improve salt tolerance in olive……………………….’
II. The description of soil salinization as a major environmental stress under the head ‘origins and consequences’ is rather short and needs to be elaborated further. It should be in sync with the broad theme of study i.e., salt tolerance in olive. Accordingly, the authors should modify it by including various natural and human-caused factors driving salinity development, global extent of salt-affected soils, effects on global food and nutrition security, various management practices, importance of salt tolerant cultivars, yield losses in olive etc.
III. Various parts of the manuscript seem to be largely disconnected from each other, and there is short of discontinuity and abrupt changes that reduce the quality of the paper. Therefore, the authors must ensure that each paragraph/section is logically connected to the previous one and forms an interesting story for the prospective readers.
IV. Under changing climatic conditions, the crops in general including olives are expected to be hit badly by a combination of abiotic stresses such as drought and salinity, waterlogging and salinity, heat and salinity. Although authors have sporadically mentioned at some places about the drought stress, the relevant insights to waterlogging and salinity stress is entirely missing. Therefore, they are advised to devote a paragraph or two wherein they must present an analysis of the combined impacts of these stresses on olive tree growth and development.
V. Under the head ‘Why olive trees’ authors seem to be reluctant in describing in some detail the varied effects of salinity stress on olive fruit yield and quality as well as oil content. Therefore, this part of manuscript is underdeveloped and needs more attention. Then, authors should present some well documented cases or data highlighting the scale of the problem in olive-growing countries and regions, and how it impacts olive industry and growers’’ livelihoods.
VI. Surprisingly, there is no mention of sodicity/alkali stress as well as sewage water/waste water on olive tree growth and productivity. This is more interesting because the effects of treated wastewaters is increasing in some olive growing countries in the Mediterranean basin. Perhaps, some studies must also have been conducted to examine the effects of seawater irrigation on olive trees. But this aspect has also not caught the attention of the authors.
VII. The references to field crops such as rice and wheat at many places throughout the manuscript are unwarranted. Instead, authors should take insights from other perennial fruit trees wherever necessary to better articulate their views and observations.
VIII. In Figure 1, authors have presented a schematic overview of salt stress effects and responses of olive trees. Here, mention of land degradations as one of the causes of salt buildup is not acceptable because salinity is a major driver of land degradation and not vice versa. Similarly, while Na and Cl uptake is mentioned as short-term response, there sequestration into vacuoles is listed as long-term response. This is a bit confusing and needs to be looked into.
IX. Authors should have tried to incorporate one or two images from their earlier research experiments on olive which illustrate the various impacts of salt stress on olive growth and development, and this would be more interesting to the prospective readers.
X. Various headings and subheadings of the manuscript are not appropriate and poorly presented. For instance, the heading ‘Plant management of soil salinisation’ and ‘Olive tree management of soil salinisation’ are absolutely vague and do not make any sense.
XI. Likewise, there are plenty of sentences in the entire manuscript which are not easily comprehensible and rather confusing. For example, in line 107-110, the statement that ‘Although there are many reviews considering the plant responses to salt stress, such as the one by Hualpa-Ramírez and coworkers [42] focused on possible biotechnological strategies to cope with climate change harsh conditions, a general overview of the main physiological and molecular mechanisms of salt tolerance will be illustrative’ is highly confusing. Similar instances include lines 112-114 (Natural salt stress is usually due to soil salinisation, characterised by the accumulation of water-soluble salts (mainly Na+ and Cl− accompanied by other ions) in the root 113 zone, which provokes a diminution of water uptake), lines 316-319 (Even though the olive tree has been described as an intermediate tolerant species to salinity compared to other fruit trees (but less resistant than barley, cotton, or sugar beet), a sustained excessive salt exposure may result in a cultivar-dependent, non-adaptive response presenting with reduction of stalk length, leaf surface, dry weight and root length), and many more. Such sentences should be critically revised to draw meaningful conclusions.
XII. Statements like ‘Two main phenotypes can be defined: halophytes (species adapted to perpetually saline conditions of 70-200 mM NaCl [77]) and glycophytes (plants requiring soils with low salt content to maintain low Na+ and Cl- levels in their above-ground tissues [78])’ are highly undesirable. Halophytes and glycophytes are not phenotypes but two distinct classes of plants based on their salt stress responses. Therefore, such sentences and statements need to be carefully looked into and modified for a better meaning.
XIII. Is there any report on trans-grafting in olive for any abiotic stress including salinity? This topic needs some attention.
XIV. Machine learning models seem to be overemphasized and should be cut in size.
XV. Overall, the English language is not satisfactory throughout and the article is replete with grammatic errors at many places. This is unwarranted, and English editing and proofreading are desirable to further improve the quality of the manuscript.
XVI. In the conclusions section of the paper, authors should present some insights on emerging frontier technologies in fruit breeding and their potential applications in olive crop including structural and functional biology, systems biology, epigenomics, etc. However, caution is needed in using these technologies and their implications for genetic diversity conservation, government regulatory issues, public opinion, etc. should be adequately captured. Do authors have any plans and suggestions for collaborative research in this direction including various disciplines.
XVII. Conclusions section is quite large and needs to be shortened to one or two paragraphs. The findings presented and discussed here may be shifted to other relevant sections of the manuscript.
Comments on the Quality of English LanguageEnglish language needs considerable improvement.
Author Response
Response to Reviewer #1 Comments
We would like to acknoledge the deep review of our manuscript performed by referee #1 since the following comments allowed us to improve the new version. Please find the detailed responses below and the corresponding corrections highlighted in red in the re-submitted file.
Comment 1
Authors are advised to modify the manuscript title as: ‘Advances in genomics to improve salt tolerance in olive…
Response:
Thank you for your comment. We have modified the title to:
Advances in genomics to improve salt tolerance in olive tree (Olea europaea subsp. europaea L.): stepping towards “smart oliviculture”
Comment 2
The description of soil salinization as a major environmental stress under the head ‘origins and consequences’ is rather short and needs to be elaborated further. It should be in sync with the broad theme of study i.e., salt tolerance in olive. Accordingly, the authors should modify it by including various natural and human-caused factors driving salinity development, global extent of salt-affected soils, effects on global food and nutrition security, various management practices, importance of salt tolerant cultivars, yield losses in olive etc.
Response:
Thank you for pointing this out. Since the review was very long, we initially decided to shorten this introduction, but we agree with the referee’s comment and we had tackled those aspects in modifications in lines 54-70 and 74 in Section 1.1. Then, the aspects of salinisation specifically regarding olive tree are complemented with lines 89-93 and 97-104 in Section 1.2. Aspects of agricultural practices in olive groves are mentioned throughout Sections 3 (including a modification in lines 339-344), 4 and 5, which make us think that it would not be suitable the duplication of this information in Introduction. Our idea of introduction is to shortly inform the about the salinisation and salt stress issues because it is a well known issue in the scientific field, and then why we have been focused on olive, which is a less common model in the field of salt stress. This is why we included Section 2 to overview the salt response and then the rest of the manuscript to gather informations about olive tree.
Comment 3
Various parts of the manuscript seem to be largely disconnected from each other, and there is short of discontinuity and abrupt changes that reduce the quality of the paper. Therefore, the authors must ensure that each paragraph/section is logically connected to the previous one and forms an interesting story for the prospective readers.
Response:
Thank you for your comment. We have carefully revised the text to avoid the disconnections and provide a more logical story line. They are small changes throughout the manuscript that would be tedious to mention here.
Comment 4
Under changing climatic conditions, the crops in general including olives are expected to be hit badly by a combination of abiotic stresses such as drought and salinity, waterlogging and salinity, heat and salinity. Although authors have sporadically mentioned at some places about the drought stress, the relevant insights to waterlogging and salinity stress is entirely missing. Therefore, they are advised to devote a paragraph or two wherein they must present an analysis of the combined impacts of these stresses on olive tree growth and development.
Response:
Thank you for highlighting this issue. Olive groves maybe subject of combined stresses with salinity regarding drought and heat, although waterlogging is very improbable since they are cultured in dry lands (and when occurs, it is transitory due to rainfall). In fact, as mentioned in lines 102-105, many olive groves have been moved to irrigated intensive groves with high water demands, which increases the salinisation issues. Probably, this is why we have not detect any study about waterlogging in olive literature (even if it is frequent in other species, for example in https://doi.org/10.1016/j.jia.2023.12.028).
In agreement with the referee’s comment, there are many reviews considering combined stresses (heat + drought, heat + salt or drought + salt) in plants, including olive (to cite a few, https://doi.org/10.1016/j.stress.2023.100319 (2024), https://doi.org/10.1016/j.stress.2023.100255 (2023), https://doi.org/10.3390/plants11212884, https://doi.org/10.3390/plants11233358, https://doi.org/10.3389/fpls.2022.918537 (2022), https://doi.org/10.1007/s00122-019-03331-2 (2019)). Hence, we have inserted a new Section 3.3 (lines 434-448) entitled “Combination of salt and drought stresses in olive tree” describing the most important findings when both stresses are concomitant in olive tree.
Comment 5
Under the head ‘Why olive trees’ authors seem to be reluctant in describing in some detail the varied effects of salinity stress on olive fruit yield and quality as well as oil content. Therefore, this part of manuscript is underdeveloped and needs more attention. Then, authors should present some well documented cases or data highlighting the scale of the problem in olive-growing countries and regions, and how it impacts olive industry and growers’’ livelihoods.
Response:
Thank you for your comment, that drove us to describe the effects of salt stress on olive trees taking into account that they were resumed in lines 339-344, at the beginning of Section 3. Moreover, we have deeply modified Section 1.2 to include the effects of moderate salinity irrigation on olive yield and olive oil (lines 97-118). The effects on economy were also mentioned in Sections 1.1 and 1.2 (lines 54-70, 74, 89-93) as a result of our answer to comment #2.
Comment 6
Surprisingly, there is no mention of sodicity/alkali stress as well as sewage water/waste water on olive tree growth and productivity. This is more interesting because the effects of treated wastewaters is increasing in some olive growing countries in the Mediterranean basin. Perhaps, some studies must also have been conducted to examine the effects of seawater irrigation on olive trees. But this aspect has also not caught the attention of the authors.
Response:
This is a good point, even though the review is not focused on the agricultural practices that can improve or worsen the soil salinistation but the molecular consequences of salt exposure. Since this comment is related to previous comment #5, the use of moderate saline water obtained from salinised groundwater or from sewage water has been described in lines 97-118in Section 1.2 in relation to the productivity of olive trees.
Regarding socidity, it is a particular aspect of salt stress that has not been specifically studied in olive trees. Accumulation and transport of Na and Cl, and the interference with K and Ca, are the main aspects treated in most plants, including olive tree, and this is why we have oriented our manuscript in this sense. We think that the fact that many reviews (to cite a few https://doi.org/10.3389/fenvs.2021.712831, https://doi.org/10.7176/JNSR/12-3-01, and the Annua Review of Plant Biology https://doi.org/10.1146/annurev-arplant-061422-104322) mention together salinity and sodicity may support our decision.
Comment 7
The references to field crops such as rice and wheat at many places throughout the manuscript are unwarranted. Instead, authors should take insights from other perennial fruit trees wherever necessary to better articulate their views and observations.
Response:
The comment of the referee is very appropriate. Section 2 in the manuscript was thought to quickly overview what is known about plant resposes to salt. Rice and other crop plants (mainly herbs) are cited there since these models have received many research efforts on this subject. Wheat, maize, sorghum, soybean and others are only mentioned in Introduction. References to works on these crop models may appear along the manuscript since the results with this plants are considered a kind of ‘gold standard’ mechanisms, but we hope that our manuscript clearly informs that our aim is not to describe salt strees in model plants but in olive tree. Salt stress studies on fruit trees are not in general as conclusive as those in other model plants; some works on fruit trees are however mentioned in the original manuscript, for example in lines 249-254.
Comment 8
In Figure 1, authors have presented a schematic overview of salt stress effects and responses of olive trees. Here, mention of land degradations as one of the causes of salt buildup is not acceptable because salinity is a major driver of land degradation and not vice versa. Similarly, while Na and Cl uptake is mentioned as short-term response, there sequestration into vacuoles is listed as long-term response. This is a bit confusing and needs to be looked into.
Response:
Thank you for the comment drove us to remove land degradation from Figure 1. Concerning sequestration into vacuoles, although it is initated after the detection of high salt, it is a sustained response during salt stress, and this is why we have preferred its inclusion as a long-term response. The same rationale could apply antioxidant defence.
Comment 9
Authors should have tried to incorporate one or two images from their earlier research experiments on olive which illustrate the various impacts of salt stress on olive growth and development, and this would be more interesting to the prospective readers.
Response:
Thank you for the idea. We have included Figure 2 that presents out not published yet results on roots of long-term salt stress in different olive cultivars. They clearly show that salt affects both senstive and intermediate-tolerant cultivars, while the effect on the salt-tolerant is not so prominent. It is then cited in lines 342, 391, 484, 510, 512 and 763.
Comment 10
Various headings and subheadings of the manuscript are not appropriate and poorly presented. For instance, the heading ‘Plant management of soil salinisation’ and ‘Olive tree management of soil salinisation’ are absolutely vague and do not make any sense.
Response:
Thank you very much for highlighting this flaw in our manuscript. We have the modified the headings mentioned by the referee and others to provide a more informative message. The modified headings, highlighted in the manuscript, are those of Section 2 (line 136), Section 2.1 (line 144), Section 3 (line 338), and Section 4 (line 502).
Comment 11.1
Likewise, there are plenty of sentences in the entire manuscript which are not easily comprehensible and rather confusing. For example, in line 107-110, the statement that ‘Although there are many reviews considering the plant responses to salt stress, such as the one by Hualpa-Ramírez and coworkers [42] focused on possible biotechnological strategies to cope with climate change harsh conditions, a general overview of the main physiological and molecular mechanisms of salt tolerance will be illustrative’ is highly confusing.
Response:
We acknowledge the referee to highlight obscure sentences that hamper the manuscript comprehension. The indicated sentence at the beginning of Section 2 has been reworded, as can be seen in lines 139-143.
Comment 11.2
Similar instances include lines 112-114 (Natural salt stress is usually due to soil salinisation, characterised by the accumulation of water-soluble salts (mainly Na+ and Cl− accompanied by other ions) in the root 113 zone, which provokes a diminution of water uptake)
Response:
The indicated sentence at the beginning of Section 2.1 has been shorten and reworded, as can be seen in lines 145-146.
Comment 11.3
lines 316-319 (Even though the olive tree has been described as an intermediate tolerant species to salinity compared to other fruit trees (but less resistant than barley, cotton, or sugar beet), a sustained excessive salt exposure may result in a cultivar-dependent, non-adaptive response presenting with reduction of stalk length, leaf surface, dry weight and root length), and many more. Such sentences should be critically revised to draw meaningful conclusions.
Response:
The indicated sentence at the beginning of Section 3 has been simplified and reworded, as can be seen in lines 337-340
As mentioned in comment #3, a deep revision of the manuscript has affected many other sentences; only the cases of significant changes were highlighted, such as lines 219-223, 401-408, 410-411, 425-431, 450-453, 472-475.
Comment 12
Statements like ‘Two main phenotypes can be defined: halophytes (species adapted to perpetually saline conditions of 70-200 mM NaCl [77]) and glycophytes (plants requiring soils with low salt content to maintain low Na+ and Cl- levels in their above-ground tissues [78])’ are highly undesirable. Halophytes and glycophytes are not phenotypes but two distinct classes of plants based on their salt stress responses. Therefore, such sentences and statements need to be carefully looked into and modified for a better meaning.
Response:
Thank you for pointing this out. We have changed phenotype by ‘category ‘ in the mentioned sentence (line 230), accoding the the descriptions in https://doi.org/10.3390/plants12122253.
Comment 13
Is there any report on trans-grafting in olive for any abiotic stress including salinity? This topic needs some attention.
Response:
We agree with the referee that this will be a very interesting issue that we have disregarded. Grafting olive cultivars on wild olive trees is a tradictional practice to generate stronger trees, but it is an abandoned practice in new olive groves. Hence we have included a paragraph in Section 3.4 (lines 456-475) that support the use of salt-tolerant rootstocks with grafted salt-sensitive scions in olive trees as a way to produce, in the short/middle-term, salt-tolerant trees.
Comment 14
Machine learning models seem to be overemphasized and should be cut in size.
Response:
We agree with the referee about that. We have then reduced Section 2.7 from 506 words to only 319 words (the reduction affected lines 303-319), but preserving most literature citations.
Comment 15
Overall, the English language is not satisfactory throughout and the article is replete with grammatic errors at many places. This is unwarranted, and English editing and proofreading are desirable to further improve the quality of the manuscript.
Response:
We are sorry about that. We made use of English editing services of MDPI for the new version of our manuscript.
Comment 16
In the conclusions section of the paper, authors should present some insights on emerging frontier technologies in fruit breeding and their potential applications in olive crop including structural and functional biology, systems biology, epigenomics, etc. However, caution is needed in using these technologies and their implications for genetic diversity conservation, government regulatory issues, public opinion, etc. should be adequately captured. Do authors have any plans and suggestions for collaborative research in this direction including various disciplines.
Response:
Thank you for pointing this out. The previous version of Conclusions was very long and the message was perhaps obscure. As the referee recommended in next comment, this Section has been shortened and we hope that the emerging possibilities of PCD-based targets, natural or induced PTMs, ammendments, use of salt-tolerant cultivars, as well as AI, might have on dealing with salt stress and soil salinisation in olive groves. Authors of this review come from very different disciplines (biochemistry, ecophysiology, genomics, metagenomics, bioinformatics, metacaspases and machine learning) and are in tight collaboration to produce an integrative analysis of salt-strees in olive tree with the aim to provide new approaches for the use of salt-tolerant olive cultivars or rootstocks.
Comment 17
Conclusions section is quite large and needs to be shortened to one or two paragraphs. The findings presented and discussed here may be shifted to other relevant sections of the manuscript._
Response:
Thank you for this suggestion. We have shorten the Conclusion section, that originally contained 924 words and now contains only two paragraphs for a total of 419 words (lines 794-828).
Reviewer 2 Report
Comments and Suggestions for Authors
The manuscript is devoted to a topical issue of modern biology – molecular mechanisms of plant salt tolerance, in particular, mechanisms of resistance to salinity of olive plants. The review corresponds to the direction of the journal Biology.
The authors logically correctly constructed the presentation – they started with the general biological problem of soil salinity for plants (Figure 1. Integrative scheme…), then they reasonably explained the importance of the salt tolerance problem for such an important agricultural oil crop as olives, which form olive groves. The authors presented an Overview of plant responses to salt exposure (Biochemical indicators of salt adaptation, Adaptive signaling to salt stress).
The main section of the review is 3. Olive tree management of soil salinisation. In it, the authors analyzed the literature on olive salt tolerance, outlined possible physiological, biochemical and genetic mechanisms by which the olive exhibits resistance to salinity. The review is decorated with 2 tables, in which the authors presented information on 89 varieties of olive trees and divided them into three groups: salt-tolerant, intermediate tolerance and salt-sensitive.
The next section is devoted to studies about salt stress genomics in olive tree – identification and study of the role of key genes of salt tolerance in olive. An important aspect of the review is the analysis of Microbiota changes in salinised soils of olive groves made by the authors, its presence is an important addition to the review – a general biological argument and the basis for publication in the journal Biology. Bioinformatics studies in olive tree concerning salt stress – this section logically completes the review. In this section, the authors presented currently known information on transcriptomic studies and sequencing of the genome of olive trees, under the influence of the salt factor.
As a request to the authors – a little shorten the Conclusion and, if possible (but it is not necessary), add to the existing figure and tables a diagram showing the biochemical and molecular mechanisms by which olive plants resist the effects of salinity.
The list of references is very large – it includes 252 sources, mostly (approximately 80%) from the last 10 years.
Author Response
Response to Reviewer #2 Comments
Thank you very much for taking the time to review this manuscript and the polite comments, and the rationale that indicates that Biology is a suitable journal for our review. Please find below our detailed responses to your suggestions.
Comment 1
A little shorten the Conclusion
Response 1
Thank you for this suggestion. We have shorten the Conclusion section, that originally contained 924 words and now contains only two paragraphs for a total of 419 words (lines 794-828).
Comment 2
If possible (but it is not necessary), add to the existing figure and tables a diagram showing the biochemical and molecular mechanisms by which olive plants resist the effects of salinity.
Response:
Thank you for pointing this improvement. We did our best to include in Figure 1 the main biochemical activities mentioned in the text. The figure legend was accordingly modified. However, we decided to exclude the molecular mechanisms for the sake of clarity.